# Breaking the Frozen Subspace: Importance Sampling for Low-Rank Optimization in LLM Pretraining

**Haochen Zhang[1], Junze Yin[1], Guanchu Wang[2], Zirui Liu[3], Lin F. Yang[4], Tianyi Zhang[1], Anshumali Shrivastava[1], Vladimir Braverman[1,5]**

[1]Rice University, [2]University of North Carolina at Charlotte, [3]University of Minnesota Twin Cities, [4]University of California, Los Angeles, [5]Johns Hopkins University

## Abstract

Low-rank optimization has emerged as a promising approach to enabling memory-efficient training of large language models (LLMs). Existing low-rank optimization methods typically project gradients onto a low-rank subspace, reducing the memory cost of storing optimizer states. A key challenge in these methods is selecting suitable subspaces to ensure an effective optimization trajectory. Most existing approaches select the dominant subspace to preserve gradient information, as this intuitively provides the best approximation. However, we find that in practice, the dominant subspace stops changing during pretraining, thereby constraining weight updates to similar subspaces. In this paper, we propose importance sampling for low-rank optimization in LLM pretraining with a provable convergence guarantee, which the dominant subspace approach does not have. Empirically, we demonstrate that our method significantly outperforms previous methods in LLM pretraining tasks.

## 1 Introduction

Large language models (LLMs), pretrained on next-token prediction tasks, achieve human-level text generation capabilities and exhibit zero-shot transferability to various downstream tasks [5]. They are also fine-tuned or aligned with human preferences to be expert in downstream tasks [66, 53]. Over the past few years, there has been rapid progress in LLM development, characterized by consistent growth in the number of trainable parameters and the scale of datasets [3, 31, 13, 1]. The parameter count in language models has increased from 100 million [56] to over a hundred billion [10]. However, despite their enhanced expressiveness, such large models demand extensive GPU memory for pretraining [52]. Thus, a critical question arises:

*How can we improve the memory efficiency of LLM pretraining?*

In LLM pretraining, Adam is commonly used as the optimizer due to its superior optimization performance. However, a key limitation of Adam is its memory requirement, as it necessitates storing two optimizer states, each consuming as much memory as the model itself. This poses a significant challenge, given the substantial memory demands of the model's parameters. To address this issue, researchers have explored low-rank optimization, where gradients are projected onto a low-rank subspace to reduce the memory consumption of optimizer states. These states are then projected back to their original size when updating the weights. For example, GaLore [79] and Q-GaLore [77] project gradients onto subspaces defined by the leading singular vectors corresponding to the largest singular values, a technique referred to as the dominant subspace. FLora [23] and GoLore [24], on the other hand, utilize unbiased random low-rank projections for gradients, employing the Johnson–Lindenstrauss transform. Grass [51] introduces sparse low-rank projections, which further

reduce the gradient memory footprint as well as the computation and communication costs compared to dense low-rank projections. Lastly, Fira [7] builds on GaLore by fully leveraging the error in gradient low-rank approximation to achieve improved performance.

These methods are powerful because: 1) the gradients of LLMs during pretraining exhibit an intrinsic low-rank structure, making them well-suited for compression using low-rank approximation, and 2) low-rank approximation can be applied not only to Adam but also to other optimizers that use state information. For instance, Adafactor [58] employs rank-1 factorization on the second moment in Adam to reduce the memory required for storing the second moment. Adam-mini [76] eliminates over 99% of the effective learning rate in the second moment of Adam while achieving performance on par with—or even better than—Adam. Additionally, [12] and [35] propose low-precision optimizers with 8-bit and 4-bit optimizer states. Low-rank optimization integrates seamlessly with these Adam variants, further highlighting its importance and underscoring why it deserves significant attention.

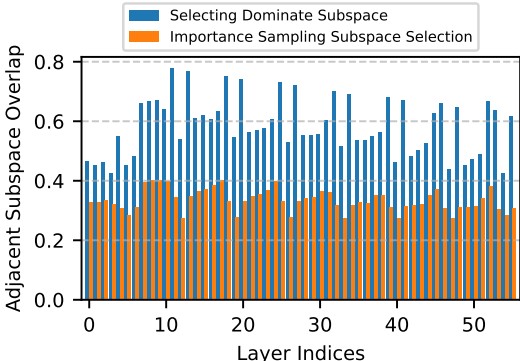

Figure 1: Adjacent subspace overlap of low-rank optimizer using difference subspace selection methods. Our importance sampling subspace selection can lower the overlap between adjacent subspaces, thus enabling better exploration of more different subspaces in the optimization trajectory.

A central question in low-rank optimization is how to maintain the performance of pretrained LLMs while using memory-efficient optimizers, as compared to full-rank optimization. One common paradigm in existing low-rank optimization methods is to update weights within the dominant subspace for a certain number of iterations and periodically update this dominant subspace. Nonetheless, the dominant subspaces of gradients in many layers stabilize almost completely after the early stages of pretraining [77]. Consequently, the weight updates during different periods predominantly remain within the same low-rank subspace, resulting in cumulative weight updates that struggle to achieve high rank. This limitation significantly hampers the language modeling capabilities of pretrained LLMs. Thus, it is natural to ask:

*Is it possible to overcome the low-rank bottleneck of existing low-rank optimization methods with minimal additional overhead?*

In this paper, we provide a positive answer to this question. We propose a novel method for subspace selection in low-rank optimization by introducing an appropriate degree of randomness in the selection process. In summary, the contributions of this study are as follows:

- We observe that highly similar adjacent subspaces in existing low-rank optimization methods diminish the diversity of weight updates, degrading the performance of pretrained LLMs.

- To address frozen dominant subspace phenomenon and the low-rank bottleneck of update in existing low-rank optimization methods, we propose an Importance **SA**mpling method for Low-**RA**nk optimization (**SARA**). This method enables low-rank optimizers to explore a broader range of subspaces in the optimization trajectory. Specifically, the low-rank subspace is spanned by $r$ singular vectors sampled from $m$ singular vectors for a gradient $G \in \mathbb{R}^{m \times n}$. Figure 1 illustrates how SARA reduces the overlap between adjacent subspaces during LLM pretraining.

- SARA can be integrated with various low-rank optimization methods, such as GaLore and Fira. It is robust to second-moment factorization and low-precision optimizer state storage. On pretraining tasks for the LLaMA model at different sizes, SARA consistently outperforms dominant subspace selection and reduces the performance gap between low-rank optimizers and full-rank Adam by up to 46.05%.

- From a theoretical aspect, We prove that SARA achieves a comparable convergence rate as GoLore [24] (Theorem 3.4, proof details are deferred to Appendix A) whereas delivering better empirical results (Section 4 and Appendix F).

**Roadmap.** In Section 2, we present the update rules of GaLore-Adam and Fira-Adam. In Section 3, we describe our methodology for using importance sampling to improve the algorithmic design of low rank optimizers—GaLore and Fira—and provide the convergence guarantee of SARA. In Section 4, we present experimental results showing that SARA consistently outperforms dominant subspace selection. In Section 5, we discuss related work. Finally, in Section 6, we conclude the paper.

## 2 Preliminaries

In this section, we present the background required for our theoretical analysis and experiments. In our experiments (Section 4), we apply SARA to two low-rank optimization methods, GaLore and Fira, both of which can be combined with stateful optimizers (e.g., Adam, Adafactor, and Adam-mini).

To ensure clarity, the update rules for GaLore-Adam and Fira-Adam are briefly explained here. For more detailed explanations, please refer to the original papers [79, 7]. In presenting these methods, we show the update rules for the weights of a single layer in the neural network. We assume that the gradient at the $t$-th iteration is a matrix $G^{(t)} \in \mathbb{R}^{m \times n}$. Without loss of generality, we assume that $m < n$ and use $r$ to represent the rank of the low-rank subspace.

**Update Rules of GaLore-Adam**  GaLore-Adam [79] requires storing an orthogonal matrix $P^{(t)} \in \mathbb{R}^{m \times r}$ that satisfies $(P^{(t)})^\top P^{(t)} = I_r$, which is updated periodically. Similar to full-rank Adam, GaLore-Adam also stores the first moment $M^{(t)} \in \mathbb{R}^{r \times n}$ and the second moment $V^{(t)} \in \mathbb{R}^{r \times n}$ for each layer's weights, and updates the weights $W^{(t)}$ as follows: $R^{(t)} = (P^{(t)})^\top G^{(t)}$, $M^{(t)} = \beta_1 M^{(t-1)} + (1 - \beta_1)R^{(t)}$, $V^{(t)} = \beta_2 V^{(t-1)} + (1 - \beta_2)R^{(t)} \circ R^{(t)}$, $N^{(t)} = \alpha P^{(t)}\frac{M^{(t)}}{\sqrt{V^{(t)}}+\xi}$, and $x^{(t)} = x^{(t-1)} - \eta \cdot N^{(t)}$. Here, $\beta_1$ and $\beta_2$ are hyperparameters for the online updates of $M^{(t)}$ and $V^{(t)}$, as in Adam. The parameter $\eta$ denotes the learning rate, and $\xi$ is a small positive constant for numerical stability.

**Update Rules of Fira-Adam**  Similar to GaLore-Adam, Fira also needs to store $M^{(t)}$, $V^{(t)}$, and $P^{(t)}$. The difference is that Fira-Adam additionally utilizes the low-rank approximation residual to update $W_l^{(t)}$. Let $S^{(t)} = (I - P^{(t)}(P^{(t)})^T)G^{(t)}$ and $x^{(t)} = x^{(t-1)} - \eta \cdot N^{(t)} - \eta \cdot \phi(S^{(t)})$. where $S^{(t)}$ represents the low-rank approximation error, $\phi(\cdot)$ represents a scaling function in Fira [7], and $N^{(t)} = \alpha P^{(t)}\frac{M^{(t)}}{\sqrt{V^{(t)}}+\xi}$ is calculated same as GaLore-Adam above.

## 3 Methodology

In this section, we first illustrate the adverse effects of a frozen dominant subspace in mini-batch gradients (Section 3.1). To address this issue, we then propose SARA for low-rank optimization (Section 3.2). Finally, we present a convergence analysis of low-rank optimization using SARA (Section 3.3).

### 3.1 Frozen Dominant Subspace of Mini-batch Gradient

[77] observes that the cosine similarity between adjacent dominant subspaces approaches 1.0 in some layers after a certain stage of LLM pretraining, indicating that the dominant subspace of the gradient almost stops evolving. We observe a similar phenomenon in our experiment as well. Figure 2 shows the average result of dominant subspace overlap in different layers across all blocks at different iterations. We notice that dominant subspace overlaps are low in all layers at the early stage of pretraining, but they increase drastically as pretraining progresses, eventually becoming stable at different levels. Among all layers, gate_proj and up_proj exhibit the highest subspace overlaps. Intuitively, a high overlap between adjacent subspaces is harmful for low-rank optimization. Considering an extreme case, when the overlap reaches 1.0, the low-rank optimizer can only change the weights within a fixed low-rank subspace. However, when the low-rank subspace shifts significantly over time, the overall weight update—formed by summing updates from various low-rank subspaces—can overcome the constraints of the low-rank bottleneck. For readability, we refer to this phenomenon as the frozen dominant subspace.

---

**Algorithm 1** Low-rank Optimization with SARA

---

1: **Input:** The $l$-th layer weight $x_l^{(t)} \in \mathbb{R}^{m_l \times n_l}$, for all $l \in [N]$. Learning rate $\eta$, scale factor $\alpha$, decay rates $\beta_1, \beta_2$, rank $r$, subspace change frequency $\tau \in \mathbb{Z}_+$, small constant for numerical stability $\xi$.
2: **Initialize:** for all $l \in [N]$ $V_l^{(0)}, M_l^{(0)} \in \mathbb{R}^{r \times n_l} \leftarrow 0$
3: **for** $t = 1 \rightarrow T$ **do**
4:     **for** $l = 1 \rightarrow N$ **do**
5:         Compute the mini-batch gradient: $G_l^{(t)} \in \mathbb{R}^{m_l \times n_l}$
6:         $P_l^{(t)} \leftarrow \text{SARA}(G_l^{(t)}, \tau)$                                       ▷ see Algorithm 2
7:         $\mathcal{S} \leftarrow \{V_l^{(t-1)}, M_l^{(t-1)}, x_l^{(t)}, P_l^{(t)}, G_l^{(t)}, \beta_1, \beta_2, \xi, \eta, \alpha\}$         ▷ input parameters
8:         $x_l^{(t)} \leftarrow \text{GALORE-ADAM}(\mathcal{S})$ or $\text{FIRA-ADAM}(\mathcal{S})$               ▷ see Section 2.
9:     **end for**
10: **end for**
11: **Return** $x^{(T)} = (x_1^{(T)}, x_2^{(T)}, \cdots, x_N^{(T)})$

---

## 3.2 SARA: Importance SAmpling for Low-RAnk Optimization

---

**Algorithm 2** SARA: Importance sampling subspace selection for low-rank optimization

---

1: **Input:** The mini-batch gradient at the iteration $t$, $G_l^{(t)} \in \mathbb{R}^{m \times n_l}$, where $l \in [N]$ denotes the layer. Subspace change frequency $\tau \in \mathbb{Z}_+$.
2: **if** $t \bmod \tau = 0$ **then**
3:     $U_l^{(t)}, S_l^{(t)}, V_l^{(t)} \leftarrow \text{SVD}(G_l^{(t)})$
4:     $\mathcal{I} \leftarrow \text{SAMPLE}([m], \text{num} = r, \text{weight} = S_l^{(t)})$
5:     $\mathcal{I} \leftarrow \text{SORT}(\mathcal{I})$
6:     $P_l^{(t)} \leftarrow U_l^{(t)}[:, \mathcal{I}]$
7: **else**
8:     $P_l^{(t)} \leftarrow P_l^{(t-1)}$                                     ▷ Reuse the previous projector
9: **end if**
10: **Return** $P_l^{(t)}$

---

To overcome the problem of the frozen dominant subspace problem, we propose SARA to construct low-rank subspace. Low-rank optimization with SARA is given in Algorithm 1. It can be seen that SARA does not change the overall structure of the original low-rank optimization algorithm but is a plug-and-play substitute for dominant subspace selection. Algorithm 2 gives the procedure of SARA. Line 4 denotes the weighted sampling without replacement. More precisely, each of the $m$ left singular vectors is equipped with a weight $\omega_i \in (0, 1)$ proportional to its corresponding singular value $S_i$,

$$\omega_i = \frac{S_i}{\sum_{j=1}^{m} S_j}.$$

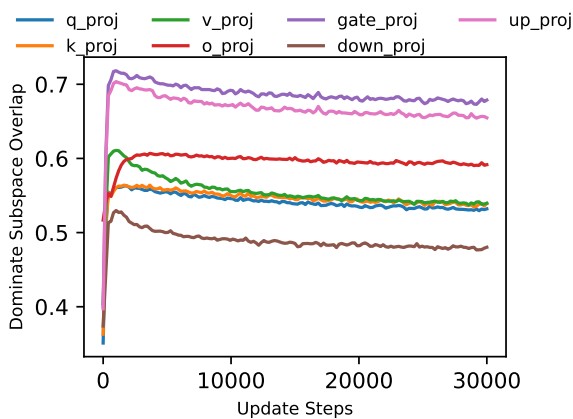

Figure 2: The average mini-batch gradient dominant subspace overlap in different linear layers over 8 blocks in the LLaMA-60M model during pretraining. We measure the overlap between adjacent subspaces every 200 iterations.

For an index set sample $\mathcal{I} = (I_1, \cdots, I_r)$, the sampling probability can be written as

$$\mathbb{P}\{(I_1, \cdots, I_r) = (i_1, \cdots, i_r)\} = \prod_{k=1}^{r} \frac{\omega_{i_k}}{1 - \omega_{i_1} - \cdots - \omega_{i_{k-1}}}$$

Line 5 sorts the sampled indices in ascending order so that the newly updated subspace basis vectors can align with optimizer states well. Line 6 constructs the orthogonal basis of the new subspace.

By using weighted sampling without replacement, we make adjacent subspaces more distinct and prevent the optimization trajectory from being trapped in overly similar subspaces during training. Another advantage of SARA is that it introduces negligible additional overhead—for example, computing an SVD on a $2048 \times 2048$ matrix takes 0.34 seconds, while sampling adds only 0.0005 seconds on average.

### 3.3 Provable Convergence of SARA

[24] points out that selecting the dominant subspace in low-rank optimization, as done in GaLore, does not always guarantee convergence to the optimal solution. Although GoLore ensures provable convergence, it does not significantly close the performance gap between GaLore-Adam and full-rank Adam in pretraining tasks, as reported in [24]. In this section, we show that SARA achieves provable convergence, representing a key advantage over GaLore and comparable to GoLore. The empirical results of SARA are shown in Section 4 and Appendix F, representing a key advantage over GaLore and GoLore.

We treat an LLM as a neural network with $N$ layers, and each layer has a weight matrix, i.e., for all $l \in [N]$, $x_l \in \mathbb{R}^{m \times n_l}$. Without loss of generality, we assume that $m \leq n_l$. In practice, most LLMs do not have biases for attention blocks and MLP blocks, and low-rank optimization is only applied to the weight matrix, but not to biases. Therefore, this abstraction is reasonable. Mathematically, our objective function is $f : \mathbb{R}^{m \times n_1} \times \mathbb{R}^{m \times n_2} \times \cdots \times \mathbb{R}^{m \times n_N} \to \mathbb{R}$. For all $x = (x_1, \ldots, x_l, \ldots, x_N), y = (y_1, \ldots, y_l, \ldots, y_N) \in \mathrm{dom}(f)$, we denote $\nabla_l f(x)$ and $\nabla_l f(y)$ as $\frac{\partial f}{\partial x_l} \in \mathbb{R}^{m \times n_l}$ and $\frac{\partial f}{\partial y_l} \in \mathbb{R}^{m \times n_l}$, respectively. Below, we use the following two assumptions similar to [24].

**Assumption 3.1** (*L*-smoothness). Let $f : \mathbb{R}^{m \times n_1} \times \mathbb{R}^{m \times n_2} \times \cdots \times \mathbb{R}^{m \times n_N} \to \mathbb{R}$ be our objective function. Let $L > 0$. For all $l \in [N]$, we let $x = (x_1, \ldots, x_l, \ldots, x_N), y = (y_1, \ldots, y_l, \ldots, y_N) \in \mathrm{dom}(f)$ be any arbitrary $N$-tuples satisfying that if $i \in [N] \setminus \{l\}$, then $x_i = y_i$. We assume $f$ is $L$-smooth that it satisfies:
$$\|\nabla_l f(x) - \nabla_l f(y)\|_F \leq L \|x_l - y_l\|_F.$$

**Assumption 3.2** (Bounded, Centered, and Independent Mini-batch Gradient Noise). Let $\nabla_l f(x^{(t)}) \in \mathbb{R}^{m \times n_l}$ be the gradient of our objective function for the $l$-th layer at the $t$-th iteration, where $t \in \mathbb{Z}_+$. Let $G_l^{(t)} \in \mathbb{R}^{m \times n_l}$ be the mini-batch gradient which is the noisy version of $\nabla_l f(x^{(t)})$. For all $l \in [N]$, we assume there exists a least upper bound $\sigma_l^2 \in \mathbb{R}$ for $\|G_l^{(t)} - \nabla_l f(x^{(t)})\|_F^2$, namely
$$\left\|G_l^{(t)} - \nabla_l f(x^{(t)})\right\|_F^2 \leq \sigma_l^2$$
and
$$\mathbb{E}\left[G_l^{(t)}\right] = \nabla_l f(x^{(t)}).$$
Furthermore, we define $\sigma^2 := \sum_{l=1}^{N} \sigma_l^2$.

To analyze the convergence of SARA, we characterize the error introduced by projecting gradients onto the sampled low-rank subspaces. Specifically, we bound the discrepancy between the original gradient and its projection under the importance sampling scheme of SARA. This projection error plays a central role in the convergence analysis, as it quantifies how well the sampled subspace preserves gradient information. Because of the page limit, we defer its proof to Appendix A.

**Lemma 3.3** (Error of SARA's Projection, see Lemma A.2 for proof). *Let $\tau$ be the update period of SARA, and $r$ be the rank of low-rank subspace in SARA. For all $i \in [m], l \in [N], k \in \mathbb{N}$, let $p_l^{(t)}(i)$ denote the probability that the $i$-th basis vector is selected for the $l$-th layer at time $t = k\tau$, and define $\delta_l^{(t)} := \min_{i \in [m]} p_l^{(t)}(i)$, $\delta := \min_{l \in [m], t \geq 0} \delta_l^{(t)}$. Let $P_l^{(t)} \in \mathbb{R}^{m \times r}$ denote the orthonormal projection matrix and let $\nabla_l f(x^{(t)}) \in \mathbb{R}^{m \times n_l}$ be the gradient matrix of the $l$-th layer at time $t$. Then, the following inequality holds:*
$$\mathbb{E}\left[\left\|\left(I - P_l^{(t)}(P_l^{(t)})^\top\right) \nabla_l f(x^{(t)})\right\|_F^2\right] \leq (1 - \delta) \cdot \mathbb{E}\left[\left\|\nabla_l f(x^{(t)})\right\|_F^2\right].$$

With the projection error bounded in Lemma 3.3, we are now equipped to analyze the convergence behavior of SARA-based low-rank optimization. The theoretical result below demonstrates that SARA achieves provable convergence at a rate comparable to prior work, while the experimental results in the following section show improved empirical performance.

**Theorem 3.4** (Convergence complexity of Low-rank MSGD with SARA, see Corollary A.6 for proof). *By Assumption 3.1-3.2, if $T \geq 2 + 128/(3\delta) + (128\sigma)^2/(9\sqrt{\delta}L\Delta)$ for $\Delta = f(x^{(0)}) - \inf_x f(x)$, we choose $\beta_1 = \left(1 + \sqrt{\frac{\delta^{3/2}\sigma^2 T}{L\Delta}}\right)^{-1}$, $\tau = \left\lceil \frac{64}{3\delta\beta_1} \right\rceil$, and $\eta = \left(4L + \sqrt{\frac{80L^2}{3\delta\beta_1^2} + \frac{80\tau^2 L^2}{3\delta}} + \sqrt{\frac{16\tau L^2}{3\beta_1}}\right)^{-1}$, low-rank MSGD-SARA with momentum re-projection converges as*

$$\frac{1}{T}\sum_{t=0}^{T-1} \mathbb{E}\left[\|\nabla f(x^{(t)})\|_2^2\right] = \mathcal{O}\left(\frac{L\Delta}{\delta^{2.5}T} + \sqrt{\frac{L\Delta\sigma^2}{\delta^{3.5}T}}\right).$$

Comparing with [24], we adopt the same hyperparameters used in their study in Theorem 3.4. When examining the convergence rate, we note that the primary distinction lies in our convergence rate depends on $\delta$ (Theorem 3.4), whereas GoLore depends on $\underline{\delta} = \frac{r}{m}$ (Theorem 3.5).

**Theorem 3.5** (Convergence of MSGD with GoLore, Corollary 3 of [24]). *Under Assumption 3.1-3.2, let every notation be defined as in Theorem 3.4, and using the same hyperparameters $\beta_1$, $\tau$, $\eta$, Let $\underline{\delta} = \frac{r}{m}$. Then, MSGD-GoLore with momentum re-projection converges as*

$$\frac{1}{T}\sum_{t=0}^{T-1} \mathbb{E}\left[\left\|\nabla f(x^{(t)})\right\|_F^2\right] = \mathcal{O}\left(\frac{L\Delta}{\underline{\delta}^{2.5}T} + \sqrt{\frac{L\Delta\sigma^2}{\underline{\delta}^{3.5}T}}\right),$$

Because SARA adopts importance sampling, we have $\delta < \underline{\delta}$. Thus, the convergence rate of MSGD-SARA is slower than MSGD-GoLore up to a constant factor. Compared to MSGD-GaLore (using dominant subspace), which does not have a provable convergence guarantee, SARA has the advantage in the theoretical convergence rate.

# 4 Experimental Results

In Section 4.1, we describe our experimental setup. In Section 4.2, we evaluate the efficacy of SARA when combined with various low-rank Adam optimizers. In Section 4.3, we show that SARA promotes subspace exploration and enables higher-rank updates. In Section 4.4, we further evaluate SARA on additional baselines and datasets.

## 4.1 Experiment Setting

In this section, we describe our experimental setup. We present the C4 dataset [57], architecture, and hyperparameters.

**Pre-training on C4 Dataset.** C4 [57], short for Colossal Clean Crawled Corpus, is a large-scale, open-source text dataset widely used in practice for pretraining transformer models such as BERT [55], T5 [68], and GPT models. C4 is also commonly used in the memory-efficient optimization community to evaluate the performance of memory-efficient optimizers [23, 79, 77, 24]. In our experiments, we pretrain LLaMA models of different sizes on the C4 dataset without data repetition, using a sufficiently large amount of data [25].

**Architecture and Hyperparameters** We evaluate the performance of different optimizers on LLaMA models with 60 million, 130 million, 350 million, and 1.1 billion parameters, using the same architecture as in [79]. For full-rank Adam, we use $\beta_1 = 0.9$, $\beta_2 = 0.999$, and a learning rate of 0.001, except for the LLaMA-60M model, where the learning rate is set to 0.0025. More detailed hyperparameters for our re-implementation are provided in Appendix B. All experiments are conducted using one GPU node with 8 Nvidia A40 GPUs.

## 4.2 Efficacy of SARA with Low-Rank Adam Optimizers

In this section, we evaluate the efficacy of SARA with different low-rank Adam optimizers across multiple model sizes.

Table 1: Validation perplexity (PPL) of LLaMA models pretrained on the C4 dataset with 60M, 130M, and 350M parameters, comparing various low-rank optimizers with and without SARA. SARA consistently reduces the PPL gap relative to full-rank Adam, demonstrating its effectiveness across different optimizer variants and model scales.

|  | **60M** | **130M** | **350M** |
|---|---|---|---|
| Full-Rank Adam | 27.71 | 23.27 | 18.21 |
| GaLore-SARA-Adam | **30.47** | **24.21** | **19.16** |
| GaLore-Adam | 31.50 | 24.88 | 19.68 |
| PPL gap reduction | 27.17% | 41.61% | 35.37% |
| Fira-SARA-Adam | **28.12** | **22.22** | **17.25** |
| Fira-Adam | 28.42 | 22.37 | 17.35 |
| PPL gap reduction | 42.25% | — | — |
| GaLore-SARA-Adafactor | **30.06** | **24.09** | **18.88** |
| GaLore-Adafactor | 31.13 | 24.79 | 19.45 |
| PPL gap reduction | 31.28% | 46.05% | 45.96% |
| GaLore-SARA-Adam-mini | **31.66** | **24.87** | **19.41** |
| GaLore-Adam-mini | 32.08 | 25.46 | 19.89 |
| PPL gap reduction | 9.61% | 26.94% | 28.57% |
| GaLore-SARA-Adam (8bit) | **30.55** | **24.67** | **18.16** |
| GaLore-Adam (8bit) | 31.62 | 25.35 | 18.63 |
| PPL gap reduction | 27.36% | 32.69% | — |
| $r/d_{model}$ | 128/256 | 256/768 | 256/1024 |
| Tokens | 1.5B | 2.2B | 6B |

**Efficacy of SARA with different low-rank Adam optimizers**    First, we evaluate the efficacy of SARA when combined with various low-rank Adam optimizers. Table 1 shows that SARA consistently outperforms dominant subspace selection. In cases where full-rank Adam achieves the lowest PPL, we also report the percentage reduction in the PPL gap achieved by SARA compared to the dominant subspace baseline. As shown in Table 1, SARA reduces the PPL gap by up to 46.05%. In scenarios where full-rank Adam does not achieve the lowest PPL, SARA still improves performance over leading singular vector selection. SARA proves effective not only with low-rank Adam variants such as GaLore-Adam and Fira-Adam, but also with optimizers that approximate second moments, e.g., GaLore-Adafactor and GaLore-Adam-mini. Results with the 8-bit optimizer further highlight the robustness of SARA under low-precision optimizer state storage.

**Scale Up to Llama-1.1B**    We also evaluate the efficacy of SARA on the pretraining of LLaMA-1.1B. Due to limited computational resources, we conduct experiments using only GaLore-Adam. As shown in Table 2, SARA remains effective on LLaMA-1.1B.

Table 2: PPL on LLaMA-1.1B pretrained with full-rank Adam, GaLore-Adam, and GaLore-SARA-Adam on the C4 dataset. Despite the larger model size, SARA continues to outperform dominant subspace selection, confirming its scalability and robustness.

|  | Full | GaLore-SARA-Adam | GaLore-Adam |
|---|---|---|---|
| 1.1B | 15.90 | **15.36** | 15.47 |
| $r/d_{model}$ | 512/2048 | 512/2048 | 512/2048 |
| Tokens | 13.4B | 13.4B | 13.4B |

### 4.3    SARA Encourages Subspace Exploration and Higher-Rank Updates

In this section, we empirically show that SARA encourages subspace exploration and enables higher-rank updates.

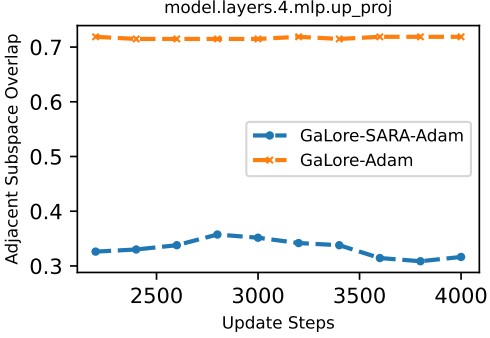 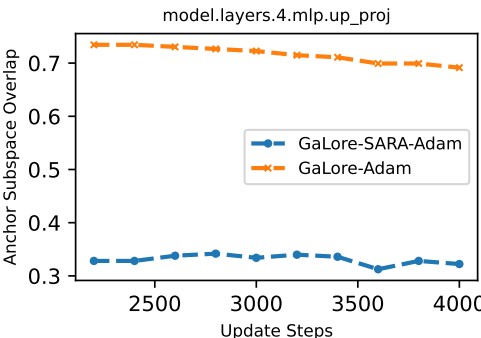

Figure 3: a). The left figure shows the overlap between adjacent subspaces in GaLore-Adam and GaLore-SARA-Adam during pretraining on the LLaMA-60M model between 2200-th and 4000-th iteration. b). The right figure takes the low-rank subspace at the 2000-th iteration as the anchor subspace, and shows the overlap between subspaces in later iterations and the anchor subspace.

**SARA encourages subspace exploration**  [77] provides an interesting observation that the similarity between adjacent subspaces in some layers gradually becomes very high during pretraining, we observe a similar phenomenon shown in Figure 2. We adopt a metric to measure overlap between two subspaces from [21]. Given two orthonormal matrices $U, V \in \mathbb{R}^{m \times r}$, we have

$$U^T U = V^T V = I_r,$$

the overlap between two subspaces spanned by $U$ and $V$ are defined as

$$\text{overlap}(U, V) = \frac{1}{r} \sum_{i=1}^{r} \|U^T V_{:,i}\|_2^2,$$

where $V_{:,i}$ denotes the $i$-th column of $V$. We adopt the above metric to show that the observation in [77] is not because of using cosine similarity as the measure, but the frozen subspace phenomenon also exists when using other metrics to measure subspace overlap (or subspace similarity).

An interesting fact is that the overlap between adjacent subspaces in GaLore-SARA-Adam is much lower than GaLore-Adam, as shown in Figure 3 (a). In Figure 3 (b), we chose the subspace at the 2000-th iteration as the anchor subspace and examined the overlap between subspaces from later iterations, specifically between the 2200-th and 4000-th iterations. We observe that the overlap between the anchor subspace and later subspaces of GaLore-SARA-Adam is lower than that of GaLore-Adam. This indicates that SARA encourages the optimization trajectory to explore more different subspaces compared to using the dominant subspace.

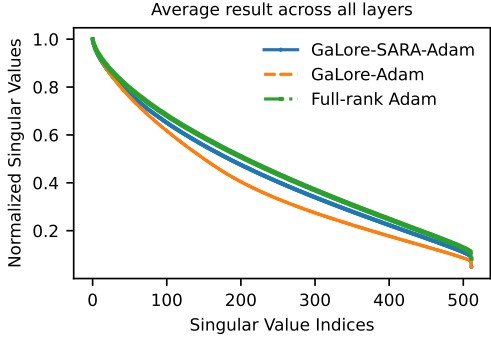

Figure 4: The average result of normalized singular values of the weight difference between the 28k-step checkpoint and 30k-step checkpoint across all layers in LLaMA-60M during pretraining.

**SARA Enables Higher-rank Update**  Figure 4 shows that the update produced by SARA in the weight matrix exhibits more evenly distributed singular values compared to the update using the dominant subspace. This suggests that SARA helps overcome the low-rank bottleneck associated with the dominant subspace approach. We observe both this higher-rank update and improved subspace exploration occurring simultaneously. We believe these two phenomena are correlated. One possible explanation is that better exploration of diverse subspaces leads to a higher-rank update.

## 4.4 Additional Baselines and Datasets

In this section, we present additional baselines (GoLore [24] and online PCA [42]) and a dataset (SlimPajama).

**More Baselines for Pretraining on C4**    To provide a more comprehensive evaluation of SARA, we conduct extensive pretraining experiments on the C4 dataset with additional baseline methods. Beyond the comparisons shown in our main results, we include two particularly relevant baselines: GoLore [24] and the online PCA approach from [42]. These baselines were selected because they are also competitive alternatives to our method. The results, presented in Table 3, reveal several important insights. First, we observe that GoLore-Adam performs reasonably well, achieving validation perplexity of 31.61 and 24.01 for the 60M and 130M parameter models respectively. However, SARA consistently outperforms GoLore by significant margins (1.14 and 0.93 perplexity points for the two model sizes), demonstrating the effectiveness of our approach in learning more efficient low-rank representations. The comparison with [42]'s online PCA method is particularly illuminating. Their approach, while computationally efficient, shows substantially higher perplexity (33.69 and 30.62) compared to both GoLore and SARA. This is mainly due to the unstable training loss curve during the online PCA method pretraining.

Table 3: Validation perplexity comparison on the C4 dataset for LLaMA models with 60M and 130M parameters using additional baselines.

| C4 Dataset (validation perplexity) | 60M (1.5B tokens) | 130M (3B tokens) |
|---|---|---|
| GoLore-Adam | 31.61 | 24.01 |
| [42] with Adam | 33.69 | 30.62 |
| GaLore-SARA-Adam | 30.47 | 23.08 |
| Full rank Adam | 27.71 | 22.19 |

**Pretraining Results on More Datasets**    To demonstrate the generalizability of SARA beyond the C4 dataset, we conduct additional pretraining experiments on the SlimPajama dataset, a carefully filtered and deduplicated subset of the Pile corpus. The results in Table 4 confirm that our method's advantages are not dataset-specific. Several interesting patterns emerge from the SlimPajama results. First, the performance gaps between methods are slightly smaller on SlimPajama compared to C4. For the 130M parameter model, SARA improves upon standard GaLore-Adam by 0.57 perplexity points (25.23 vs 25.80), while trailing full-rank Adam by only 0.23 points. This suggests that our method may be particularly effective on higher-quality, deduplicated datasets like SlimPajama. We also note that the absolute perplexity values are slightly lower on SlimPajama than on C4 for comparable model sizes, which may reflect the dataset's more careful curation. This makes the strong performance of SARA even more noteworthy, as it demonstrates effectiveness across different difficulty levels.

Table 4: Validation perplexity on the SlimPajama dataset for LLaMA models with 60M and 130M parameters.

| SlimPajama (validation perplexity) | 60M | 130M |
|---|---|---|
| Full rank Adam | 27.79 | 25.00 |
| GaLore-Adam | 31.76 | 25.80 |
| GaLore-SARA-Adam | 30.79 | 25.23 |

Additional experiments on high-rank updates, anchor similarity, and adjacent overlap can be found in Appendix F.1, F.2, and F.3, respectively.

## 5    Related Work

**Memory Efficient Parametrization.**    LoRA [26] can be seen as a memory-efficient parametrization of weights in LLMs and is widely used in fine-tuning. LoRA's bottleneck lies in its low-rank structure, which impedes its expressiveness. COLA [67], Delta-LoRA [81], and PLoRA [47] propose to

increase the rank and improve the performance of LoRA. ReLoRA [40] and SLTrain [22] extend LoRA to pre-training tasks by merging and resetting adapters, and adopting low-rank plus sparse parameterization, respectively. MoRA [32] alleviates the shortcoming of the low-rank disadvantage of LoRA by sharing the same trainable parameters to achieve a higher-rank update. Additionally, [45] analyzes the sparsity-based parameter-efficient fine-tuning (SPEFT) for LLMs, which is an alternative method of LoRA. [54] designs a novel method for memory-efficient fine-tuning for LLMs. It has been shown that it can outperform LoRA and full-parameter training in many cases. Similarly, [28] proposes another novel fine-tuning method called Half Fine-Tuning (HFT), which can mitigate "catastrophic forgetting" in LLMs during sequential training and instruction tuning. Finally, [36] also proposes a novel memory-efficient fine-tuning method by strategically selecting layers to update based on outlier statistics. Our paper also considers the memory and convergence behavior, but the difference is that we mainly focus on the LLM pretraining instead of fine-tuning.

**Memory Efficient Optimizer.**   One way to achieve memory-efficient optimization is by using memory-efficient optimizers, which primarily aim to reduce the memory cost of optimizer states in Adam [33]. A series of works [58, 76, 46, 80] factorize the second moment in Adam. Quantizing optimizer states and storing them in low-precision formats has also proven successful [35, 12]. Another line of work focuses on gradient compression methods. GaLore [79] and Q-GaLore [77] use SVD to apply dense low-rank projections to gradients. FLora [23] and GoLore [24] adopt random projection, while Grass [51] employs sparse low-rank projection to gradients.

**Subspace Learning.**   Existing studies provide sophisticated analyses of various subspace learning algorithms [11, 34, 29]. [21] claims that gradient descent primarily occurs in the dominant subspace, which is spanned by the top eigenvectors of the Hessian. In contrast, [60] argues that, due to noise in SGD, the alignment between the gradient and the dominant subspace is spurious, and learning does not occur in the dominant subspace but rather in its orthogonal complement, i.e., the bulk subspace. Intuitively, our findings align with those of [60], suggesting that selecting basis vectors based on specific sampling probabilities can enhance the performance of LLMs during pre-training.

# 6   Conclusion

In this paper, we propose SARA for low-rank optimization in LLM pretraining. The motivation is to find an effective subspace selection method to overcome the low-rank bottleneck caused by the frozen dominant subspace in low-rank optimization. SARA samples singular vectors of mini-batch gradients with probabilities proportional to their singular values, this enables optimization trajectory to explore more different subspaces. Theoretically, in Theorem 3.4, we show that GaLore-SARA-MSGD achieves comparison convergence rate as GoLore-MSGD, which is $\mathcal{O}\left(\frac{L\Delta}{\delta^{2.5}T} + \sqrt{\frac{L\Delta\sigma^2}{\delta^{3.5}T}}\right)$.

Empirically, we find that SARA improves the language modeling capability of pretrained models compared to using the dominant subspace, as verified by experiments involving SARA and dominant subspace selection with multiple low-rank optimizers.

# Acknowledgment

Lin F. Yang is supported in part by NSF Grant 2221871 and an Amazon Faculty Award. Vladimir Braverman is supported in part by NSF Grant CNS 2528780.

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

# Appendix

## A Proof Details for SARA

**Lemma A.1** (Gradient connections, Lemma 2 in [24]). *Let $\nabla_\ell f(x)$ denote the gradient with respect to the $\ell$-th layer parameters at point $x$, and suppose $x^{(t)}$ denotes the parameters at iteration $t$. Then for any integers $t \geq 0$ and $\tau > 0$, we have*

*Then, for all $t, \tau > 0$, we have*

$$\|\nabla_\ell f(\mathbf{x}^{(t)})\|_F^2 \leq \frac{2}{\tau} \sum_{r=0}^{\tau-1} \|\nabla_\ell f(\mathbf{x}^{(t+r)})\|_F^2 + (\tau - 1) \sum_{r=0}^{\tau-2} \left\|\nabla_\ell f(\mathbf{x}^{(t+r+1)}) - \nabla_\ell f(\mathbf{x}^{(t+r)})\right\|_F^2.$$

**Lemma A.2** (Error of SARA's Projection). *Let $\tau$ be the update period of SARA, and $r$ be the rank of low-rank subspace in SARA. For all $i \in [m], l \in [N], k \in \mathbb{N}$, let $p_l^{(t)}(i)$ denote the probability that the $i$-th basis vector is selected for the $l$-th layer at time $t = k\tau$, and define*

$$\delta_l^{(t)} := \min_{i \in [m]} p_l^{(t)}(i), \quad \delta := \min_{l \in [m], t \geq 0} \delta_l^{(t)}$$

*Let $P_l^{(t)} \in \mathbb{R}^{m \times r}$ denote the orthonormal projection matrix and let $\nabla_l f(x^{(t)}) \in \mathbb{R}^{m \times n_l}$ be the gradient matrix of the $l$-th layer at time $t$. Then, the following inequality holds:*

$$\mathbb{E}\left[\left\|\left(I - P_l^{(t)}(P_l^{(t)})^\top\right) \nabla_l f(x^{(t)})\right\|_F^2\right] \leq (1 - \delta) \cdot \mathbb{E}\left[\left\|\nabla_l f(x^{(t)})\right\|_F^2\right].$$

*Proof.* We analyze the expected projection residual:

$$\mathbb{E}\left[\left\|\left(I - P_l^{(t)}(P_l^{(t)})^\top\right) \nabla_l f(x^{(t)})\right\|_F^2\right]$$

$$= \mathbb{E}_{\nabla_l f(x^{(t)})}\left[\mathbb{E}_{P_l^{(t)}}\left[\left\|\left(I - P_l^{(t)}(P_l^{(t)})^\top\right) \nabla_l f(x^{(t)})\right\|_F^2 \,\middle|\, \nabla_l f(x^{(t)})\right]\right]$$

$$= \mathbb{E}_{\nabla_l f(x^{(t)})}\left[\mathrm{tr}\left(\mathbb{E}_{P_l^{(t)}}\left[\left(I - P_l^{(t)}(P_l^{(t)})^\top\right)^2\right] \cdot \nabla_l f(x^{(t)})\nabla_l f(x^{(t)})^\top\right)\right].$$

Let $\{U_j\}_{j=1}^m$ be a fixed orthonormal basis for $\mathbb{R}^m$, and define the indicator variable $\mathbf{1}_{\{j\}}$ to denote whether $U_j$ is selected. Then,

$$\mathbb{E}_{P_l^{(t)}}\left[I - P_l^{(t)}(P_l^{(t)})^\top\right] = \sum_{j=1}^m (1 - \mathbb{E}[\mathbf{1}_{\{j\}}])U_j U_j^\top.$$

Therefore,

$$\mathbb{E}\left[\left\|\left(I - P_l^{(t)}(P_l^{(t)})^\top\right) \nabla_l f(x^{(t)})\right\|_F^2\right] = \mathbb{E}\left[\mathrm{tr}\left(\sum_{j=1}^m (1 - \mathbb{E}[\mathbf{1}_{\{j\}}])U_j U_j^\top \cdot \nabla_l f(x^{(t)})\nabla_l f(x^{(t)})^\top\right)\right]$$

$$= \sum_{j=1}^m (1 - \mathbb{E}[\mathbf{1}_{\{j\}}]) \cdot \mathbb{E}\left[\left\|U_j^\top \nabla_l f(x^{(t)})\right\|_2^2\right]$$

$$\leq (1 - \min_j \mathbb{E}[\mathbf{1}_{\{j\}}]) \cdot \mathbb{E}\left[\left\|\nabla_l f(x^{(t)})\right\|_F^2\right]$$

$$= (1 - \delta_l^{(t)}) \cdot \mathbb{E}\left[\left\|\nabla_l f(x^{(t)})\right\|_F^2\right]$$

$$\leq (1 - \delta) \cdot \mathbb{E}\left[\left\|\nabla_l f(x^{(t)})\right\|_F^2\right]. \qquad \square$$

**Lemma A.3** (Momentum Contraction of SARA). *For SARA with momentum re-projection, for all $l \in [N]$, we have the inequalities below hold.*

- **Part 1** *(t = 0)*.

$$\mathbb{E}\left[\left\|\widetilde{M}_l^{(0)} - \nabla_l f(x^{(0)})\right\|_F^2\right] \le \left(1 - (2\beta_1 - \beta_1^2)\delta\right)\mathbb{E}\left[\left\|\nabla_l f(x^{(0)})\right\|_F^2\right] + \beta_1^2\sigma_l^2.$$

- **Part 2** *(t = kτ, k ∈ ℕ)*.

$$\mathbb{E}\left[\left\|\widetilde{M}_l^{(t)} - \nabla_l f(x^{(t)})\right\|_F^2\right] - \left(1 - \left(1 - \frac{\delta}{4}\right)\beta_1\right)\mathbb{E}\left[\left\|\widetilde{M}_l^{(t-1)} - \nabla_l f(x^{(t-1)})\right\|_F^2\right]$$
$$\le \frac{2(1-\delta)}{\tau}\sum_{r=0}^{\tau-1}\mathbb{E}\left[\left\|\nabla_l f(x^{(k\tau+r)})\right\|_F^2\right] + \frac{5(1-\beta_1)}{\beta_1\delta}\mathbb{E}\left[\left\|\nabla_l f(x^{(t)}) - \nabla_l f(x^{(t-1)})\right\|_F^2\right]$$
$$+ (\tau-1)(1-\delta)\sum_{r=0}^{\tau-2}\mathbb{E}\left[\left\|\nabla_l f(x^{(k\tau+r+1)}) - \nabla_l f(x^{(k\tau+r)})\right\|_F^2\right] + \beta_1^2\sigma_l^2.$$

- **Part 3** *(t = kτ + r, 1 ≤ r ≤ τ − 1)*.

$$\mathbb{E}\left[\left\|\widetilde{M}_l^{(t)} - \nabla_l f(x^{(t)})\right\|_F^2\right] - \left(1 - \left(1 - \frac{\delta}{4}\right)\beta_1\right)\mathbb{E}\left[\left\|\widetilde{M}_l^{(t-1)} - \nabla_l f(x^{(t-1)})\right\|_F^2\right]$$
$$\le \left(1 - \frac{\delta}{2}\right)\beta_1\mathbb{E}\left[\left\|\nabla_l f(x^{(t)})\right\|_F^2\right] + \frac{5(1-\beta_1)}{\beta_1\delta}\mathbb{E}\left[\left\|\nabla_l f(x^{(t)}) - \nabla_l f(x^{(t-1)})\right\|_F^2\right]$$
$$+ \frac{10r\beta_1}{\delta}\sum_{i=1}^{r}\mathbb{E}\left[\left\|\nabla_l f(x^{(k\tau+i)}) - \nabla_l f(x^{(k\tau+i-1)})\right\|_F^2\right] + \beta_1^2\sigma_l^2.$$

*Proof.* **Proof of Part 1.**

Suppose $t = 0$. By definition of the momentum estimator $\widetilde{M}_l^{(0)} = \beta_1 P_l^{(0)}(P_l^{(0)})^\top G_l^{(0)}$, we decompose the error as:

$$\mathbb{E}\left[\left\|\widetilde{M}_l^{(0)} - \nabla_l f(x^{(0)})\right\|_F^2\right] = \mathbb{E}\left[\left\|\beta_1 P_l^{(0)}(P_l^{(0)})^\top\left(G_l^{(0)} - \nabla_l f(x^{(0)})\right)\right\|_F^2\right]$$
$$+ \mathbb{E}\left[\left\|\left(\beta_1 P_l^{(0)}(P_l^{(0)})^\top - I\right)\nabla_l f(x^{(0)})\right\|_F^2\right].$$

The first term is bounded using the assumption that $G_l^{(0)}$ is an unbiased estimator with variance $\sigma_l^2$:

$$\mathbb{E}\left[\left\|\beta_1 P_l^{(0)}(P_l^{(0)})^\top\left(G_l^{(0)} - \nabla_l f(x^{(0)})\right)\right\|_F^2\right] \le \beta_1^2\sigma_l^2.$$

For the second term, we apply Lemma A.2:

$$\mathbb{E}\left[\left\|\left(I - \beta_1 P_l^{(0)}(P_l^{(0)})^\top\right)\nabla_l f(x^{(0)})\right\|_F^2\right] \le \left(1 - (2\beta_1 - \beta_1^2)\delta\right)\mathbb{E}\left[\left\|\nabla_l f(x^{(0)})\right\|_F^2\right].$$

Combining both bounds:

$$\mathbb{E}\left[\left\|\widetilde{M}_l^{(0)} - \nabla_l f(x^{(0)})\right\|_F^2\right] \le \left(1 - (2\beta_1 - \beta_1^2)\delta\right)\mathbb{E}\left[\left\|\nabla_l f(x^{(0)})\right\|_F^2\right] + \beta_1^2\sigma_l^2.$$

**Proof of Part 2.**

Suppose $t = k\tau$. At the projection step, we have the update rule:

$$\widetilde{M}_l^{(t)} = P_l^{(t)}(P_l^{(t)})^\top\left((1 - \beta_1)\widetilde{M}_l^{(t-1)} + \beta_1 G_l^{(t)}\right).$$

Then:

$$\mathbb{E}\left[\left\|\widetilde{M}_l^{(t)} - \nabla_l f(x^{(t)})\right\|_F^2\right] = \mathbb{E}\left[\left\|P_l^{(t)}(P_l^{(t)})^\top\left((1 - \beta_1)\widetilde{M}_l^{(t-1)} + \beta_1 G_l^{(t)} - \nabla_l f(x^{(t)})\right)\right. \quad (1)$$

$$-\left(I - P_l^{(t)}(P_l^{(t)})^\top\right)\nabla_l f(x^{(t)})\bigg\|_F^2\bigg] \tag{2}$$

$$= \mathbb{E}\left[\left\|P_l^{(t)}(P_l^{(t)})^\top\left((1-\beta_1)\widetilde{M}_l^{(t-1)} + \beta_1 G_l^{(t)} - \nabla_l f(x^{(t)})\right)\right\|_F^2\right]$$

$$+ \mathbb{E}\left[\left\|\left(I - P_l^{(t)}(P_l^{(t)})^\top\right)\nabla_l f(x^{(t)})\right\|_F^2\right]. \tag{3}$$

The second equality is because of the Pythagorean Theorem.

Applying Lemma A.2 to the second term:

$$\mathbb{E}\left[\left\|\left(I - P_l^{(t)}(P_l^{(t)})^\top\right)\nabla_l f(x^{(t)})\right\|_F^2\right] \leq (1-\delta)\mathbb{E}\left[\left\|\nabla_l f(x^{(t)})\right\|_F^2\right].$$

The first term is bounded as below:

$$\mathbb{E}\left[\left\|P_l^{(t)}(P_l^{(t)})^\top\left((1-\beta_1)\widetilde{M}_l^{(t-1)} + \beta_1 G_l^{(t)} - \nabla_l f(x^{(t)})\right)\right\|_F^2\right]$$

$$\leq \mathbb{E}\left[\left\|(1-\beta_1)\widetilde{M}_l^{(t-1)} + \beta_1 G_l^{(t)} - \nabla_l f(x^{(t)})\right\|_F^2\right]$$

$$= \mathbb{E}\left[\left\|(1-\beta_1)(\widetilde{M}_l^{(t-1)} - \nabla_l f(x^{(t)})) + \beta_1(G_l^{(t)} - \nabla_l f(x^{(t)}))\right\|_F^2\right]$$

$$\leq (1-\beta_1)\mathbb{E}\left[\left\|\widetilde{M}_l^{(t-1)} - \nabla_l f(x^{(t)})\right\|_F^2\right] + \beta_1\mathbb{E}\left[\left\|G_l^{(t)} - \nabla_l f(x^{(t)})\right\|_F^2\right]$$

$$\leq (1-\beta_1)\mathbb{E}\left[\left\|\widetilde{M}_l^{(t-1)} - \nabla_l f(x^{(t)})\right\|_F^2\right] + \beta_1\sigma_l^2, \tag{4}$$

where the first inequality is because of $\left\|P_l^{(t)}(P_l^{(t)})^T\right\|_2 = 1$, the second inequality is because of the independent noise of mini-batch gradient, the third inequality is because of the bounded noise assumption.

To bound $\mathbb{E}\left[\left\|\widetilde{M}_l^{(t-1)} - \nabla_l f(x^{(t)})\right\|_F^2\right]$, we apply Young's inequality:

$$\mathbb{E}\left[\left\|\widetilde{M}_l^{(t-1)} - \nabla_l f(x^{(t)})\right\|_F^2\right]$$

$$\leq \left(1 + \frac{\delta\beta_1}{4}\right)\mathbb{E}\left[\left\|\widetilde{M}_l^{(t-1)} - \nabla_l f(x^{(t-1)})\right\|_F^2\right] + \left(1 + \frac{4}{\delta\beta_1}\right)\mathbb{E}\left[\left\|\nabla_l f(x^{(t)}) - \nabla_l f(x^{(t-1)})\right\|_F^2\right] \tag{5}$$

So far, we can have the upper bound for the first term in Eq. (4). By applying Eq. (5), Eq. (4), and Lemma A.1 which naturally holds in our problem setting, we have shown **Part 2**.

**Proof of Part 3.**

Suppose $t = k\tau + r, 1 \leq r \leq \tau - 1$). In this case, $P_l^{(t)} = P_l^{(k\tau)}$ is reused. The update becomes:

$$\widetilde{M}_l^{(t)} = (1-\beta_1)\widetilde{M}_l^{(t-1)} + \beta_1 P_l^{(t)}(P_l^{(t)})^\top G_l^{(t)}.$$

Using the standard decomposition:

$$\widetilde{M}_l^{(t)} - \nabla_l f(x^{(t)}) = (1-\beta_1)(\widetilde{M}_l^{(t-1)} - \nabla_l f(x^{(t)})) + \beta_1(P_l^{(t)}(P_l^{(t)})^\top - I)\nabla_l f(x^{(t)})$$

$$+ \beta_1 P_l^{(t)}(P_l^{(t)})^\top(G_l^{(t)} - \nabla_l f(x^{(t)})).$$

By unbiasedness, we have

$$\mathbb{E}\left[\left\|\widetilde{M}_l^{(t)} - \nabla_l f(x^{(t)})\right\|_F^2\right]$$

$$= \mathbb{E}\left[\left\|(1-\beta_1)(\widetilde{M}_l^{(t-1)} - \nabla_l f(x^{(t)})) + \beta_1(P_l^{(t)}(P_l^{(t)})^\top - I)\nabla_l f(x^{(t)})\right\|_F^2\right]$$

$$+ \mathbb{E}\left[\left\|\beta_1 P_l^{(t)}(P_l^{(t)})^\top(G_l^{(t)} - \nabla_l f(x^{(t)}))\right\|_F^2\right].$$

Recall that by $\left\|P_l^{(t)}(P_l^{(t)})^T\right\|_2 = 1$ and the noise assumption, so we have

$$\mathbb{E}\left[\left\|\beta_1 P_l^{(t)}(P_l^{(t)})^\top(G_l^{(t)} - \nabla_l f(x^{(t)}))\right\|_F^2\right] \leq \beta_1^2 \sigma_l^2.$$

By applying Jensen's inequality, we have

$$\mathbb{E}\left[\left\|\widetilde{M}_l^{(t)} - \nabla_l f(x^{(t)})\right\|_F^2\right] \leq (1-\beta_1)\mathbb{E}\left[\left\|\widetilde{M}_l^{(t-1)} - \nabla_l f(x^{(t)})\right\|_F^2\right]$$

$$+ \beta_1 \mathbb{E}\left[\left\|(P_l^{(t)}(P_l^{(t)})^\top - I)\nabla_l f(x^{(t)})\right\|_F^2\right] + \beta_1^2 \sigma_l^2.$$

We now bound the projection error using Lemma A.2:

$$\mathbb{E}\left[\left\|(P_l^{(t)}(P_l^{(t)})^\top - I)\nabla_l f(x^{(t)})\right\|_F^2\right] \leq \left(1 + \frac{\delta}{4}\right)\mathbb{E}\left[\left\|(P_l^{(t)}(P_l^{(t)})^\top - I)\nabla_l f(x^{(k\tau)})\right\|_F^2\right]$$

$$+ \left(1 + \frac{4}{\delta}\right)\mathbb{E}\left[\left\|\nabla_l f(x^{(t)}) - \nabla_l f(x^{(k\tau)})\right\|_F^2\right]$$

$$\leq \left(1 - \frac{3\delta}{4}\right)\mathbb{E}\left[\left\|\nabla_l f(x^{(k\tau)})\right\|_F^2\right]$$

$$+ \left(1 + \frac{4}{\delta}\right)\mathbb{E}\left[\left\|\nabla_l f(x^{(t)}) - \nabla_l f(x^{(k\tau)})\right\|_F^2\right].$$

The first equality is because of Young's inequality, the second inequality is because of Lemma A.2.

This concludes the bound. The final contraction inequality is then followed by applying this bound and collecting terms. □

For **Part 3** of Lemma A.3, we get exactly the same bound for our SARA compared with their GoLore.

Though our Momentum Contraction result is a little worse than the one in [24], we can still get the same result for Momentum Error Bound, as shown in Lemma A.4.

**Lemma A.4** (Momentum Error Bound of MSGD with SARA). *Define*

$$\sigma^2 = \sum_{l \in [N]} \sigma_l^2$$

*Then we have*

$$\sum_{t=0}^{K\tau-1} \mathbb{E}\left[\left\|\widetilde{M}^{(t)} - \nabla f(x^{(t)})\right\|_F^2\right]$$

$$\leq \left(\frac{5(1-\beta_1)}{(1-\delta/4)\delta\beta_1^2} + \frac{5\tau(1-\tau)}{(1-\delta/4)\delta} + \frac{\tau-1}{(1-\delta/4)\beta}\right) L^2 \sum_{t=0}^{K\tau-2} \mathbb{E}\left[\left\|x^{(t+1)} - x^{(t)}\right\|_F^2\right]$$

$$+ \left(\frac{1-\delta/2}{1-\delta/4} + \frac{2}{(1-\delta/4)\tau\beta_1}\right) \sum_{t=0}^{K\tau-2} \mathbb{E}\left[\left\|\nabla f(x^{(t)})\right\|_F^2\right] + \frac{K\tau\beta_1\sigma^2}{1-\delta/4}$$

*Proof.* First we apply summation to **Part 3** of Lemma A.3 as follows:

$$\sum_{t=k\tau+1}^{(k+1)\tau-1} \mathbb{E}\left[\left\|\widetilde{M}_l^{(t)} - \nabla_l f(x^{(t)})\right\|_F^2\right] - \left(1 - (1-\frac{\delta}{4})\beta_1\right)\sum_{t=k\tau+1}^{(k+1)\tau-1} \mathbb{E}\left[\left\|\widetilde{M}_l^{(t-1)} - \nabla_l f(x^{(t-1)})\right\|_F^2\right]$$

$$\leq \left(1 - \frac{\delta}{2}\right)\beta_1 \sum_{t=k\tau+1}^{(k+1)\tau-1} \mathbb{E}\left[\left\|\nabla_l f(x^{(t)})\right\|_F^2\right]$$

$$+ \frac{5(1-\beta_1)}{\beta_1\delta} \sum_{t=k\tau+1}^{(k+1)\tau-1} \mathbb{E}\left[\left\|\nabla_l f(x^{(t)}) - \nabla_l f(x^{(t-1)})\right\|_F^2\right]$$

$$+ \frac{10\beta_1}{\delta} \sum_{r=1}^{\tau-1} r \sum_{i=1}^{r} \mathbb{E}\left[\left\|\nabla_l f(x^{(k\tau+i)}) - \nabla_l f(x^{(k\tau+i-1)})\right\|_F^2\right]$$

$$+ \beta_1^2 \sigma_l^2(\tau-1)$$

$$\leq \left(1 - \frac{\delta}{2}\right)\beta_1 \sum_{t=k\tau+1}^{(k+1)\tau-1} \mathbb{E}\left[\left\|\nabla_l f(x^{(t)})\right\|_F^2\right]$$

$$+ \left[\frac{5(1-\beta_1)}{\beta_1\delta} + \frac{5\beta_1\tau(\tau-1)}{\delta}\right] \sum_{t=k\tau}^{(k+1)\tau-2} \mathbb{E}\left[\left\|\nabla_l f(x^{(t+1)}) - \nabla_l f(x^{(t)})\right\|_F^2\right]$$

$$+ \beta_1^2 \sigma_l^2(\tau-1) \tag{6}$$

Then add Eq. (6) and **Part 2** of Lemma A.3 together, we have

$$\sum_{t=k\tau}^{(k+1)\tau-1} \mathbb{E}\left[\left\|\widetilde{M}_l^{(t)} - \nabla_l f(x^{(t)})\right\|_F^2\right] - \left(1 - (1 - \frac{\delta}{4})\beta_1\right) \sum_{t=k\tau}^{(k+1)\tau-1} \mathbb{E}\left[\left\|\widetilde{M}_l^{(t-1)} - \nabla_l f(x^{(t-1)})\right\|_F^2\right]$$

$$\leq \left[\left(1 - \frac{\delta}{2}\right)\beta_1 + \frac{2(1-\delta)}{\tau}\right] \sum_{t=k\tau}^{(k+1)\tau-1} \mathbb{E}\left[\left\|\nabla_l f(x^{(t)})\right\|_F^2\right]$$

$$+ \left[\frac{5(1-\beta_1)}{\beta_1\delta} + \frac{5\beta_1\tau(\tau-1)}{\delta} + (\tau-1)(1-\delta)\right] \sum_{t=k\tau}^{(k+1)\tau-2} \mathbb{E}\left[\left\|\nabla_l f(x^{(t+1)}) - \nabla_l f(x^{(t)})\right\|_F^2\right]$$

$$+ \beta_1^2 \sigma_l^2 \tau \tag{7}$$

Then applying summation over $k$ from 0 to $K$ and summation over all $l \in [N]$ gives us the desired result. $\qquad\square$

So far, we have the comparable result of upper bound of momentum error, in the next step, apply the same proof procedure as in [24] give us the convergence of low-rank MSGD with SARA.

**Theorem A.5.** *Under Assumptions 1–3, if hyperparameters*

$$0 < \beta_1 \leq 1, \quad \tau \geq \frac{64}{3\beta_1\delta}, \quad 0 < \eta \leq \min\left\{\frac{1}{4L}, \sqrt{\frac{3\delta\beta_1^2}{80L^2}}, \sqrt{\frac{3\delta}{80\tau^2 L^2}}, \sqrt{\frac{3\beta_1}{16\tau L^2}}\right\},$$

*MSGD-SARA with momentum re-projection converges as*

$$\frac{1}{K\tau} \sum_{t=0}^{K\tau-1} \mathbb{E}\left[\|\nabla f(x^{(t)})\|_2^2\right] \leq \frac{16\Delta}{\delta\eta K\tau} + \frac{32\beta_1\sigma^2}{3\delta}$$

*for any $K \geq 1$, where $\Delta = f(x^{(0)}) - \inf_x f(x)$.*

**Corollary A.6** (Convergence complexity of Low-rank MSGD with SARA). *Under Assumption 3.1-3.2, if $T \geq 2 + 128/(3\delta) + (128\sigma)^2/(9\sqrt{\delta}L\Delta)$ and we choose*

$$\beta_1 = \left(1 + \sqrt{\frac{\delta^{3/2}\sigma^2 T}{L\Delta}}\right)^{-1}, \quad \tau = \left\lceil\frac{64}{3\delta\beta_1}\right\rceil, \quad \eta = \left(4L + \sqrt{\frac{80L^2}{3\delta\beta_1^2} + \frac{80\tau^2 L^2}{3\delta}} + \sqrt{\frac{16\tau L^2}{3\beta_1}}\right)^{-1},$$

*low-rank MSGD-SARA with momentum re-projection converges as*

$$\frac{1}{T} \sum_{t=0}^{T-1} \mathbb{E}\left[\|\nabla f(x^{(t)})\|_2^2\right] = \mathcal{O}\left(\frac{L\Delta}{\delta^{2.5}T} + \sqrt{\frac{L\Delta\sigma^2}{\delta^{3.5}T}}\right),$$

*where $\Delta = f(x^{(0)}) - \inf_x f(x)$.*

## B Experiment Implementation

To enable re-implementation, we provide hyperparameters herein. For fair comparison, we adopt the same hyperparameters for dominant subspace selection and SARA. Hyperparameters for GaLore with dominant subspace selection and SARA are shown in Table 5. Hyperparameters for Fira with dominant subspace selection and SARA are shown in Table 6.

Table 5: Hyperparameters for experiments with GaLore

| Name | Values |
|---|---|
| Batch Size | 512 |
| Maximum Sequence Length | 512 |
| Warmup Steps | 1000 for 60m, 2000 for 130m, 6000 for 350m, 10000 for 1.1B |
| Rank | 128 for 60m, 256 for 130m, 256 for 350m, 512 for 1.1B |
| Weight Decay | 0 |
| Learning Rate | 0.01 |
| Scheduler | Cosine |
| Optimizer Specific Parameters | Adam: $\beta_1 = 0.9$, $\beta_2 = 0.999$ |
| | Adafactor: $\beta_1 = 0.9$, $\beta_2(t) = 1 - t^{-0.8}$ |
| | Adam-mini: $\beta_1 = 0.9$, $\beta_2 = 0.95$ |

Table 6: Hyperparameters for experiments with Fira

| Name | Values |
|---|---|
| batch Size | 512 |
| Maximum sequence length | 512 |
| Warmup Steps | 1000 for 60m, 2000 for 130m, 6000 for 350m |
| rank | 128 for 60m, 256 for 130m, 256 for 350m |
| Weight Decay | 0 |
| Learning Rate | 0.01 |
| Scheduler | Cosine |
| optimizer specific parameters | Adam: $\beta_1 = 0.9$, $\beta_2 = 0.999$ |

## C More Related Work

**Large language models** Many works study the LLM from other aspects. For example, low rank approximation [20, 63] can also be applied to improve the computational complexity of (masked) attention approximation [43, 8]. Gaussian kernel density estimation is closely related to attention optimization, and similar to our work, [41] uses importance sampling to study the dynamic maintenance of KDE data structures. [16, 59, 62, 15, 18, 64, 39] analyze the attention regression problems. [9] study the computational limits of Mamba. [61] investigates the expressibility of polynomial attention. [17] applies the sketching technique to develop the decentralized large language model. [4] studies the attention optimization without requiring strict assumptions. [73] focuses on efficiently aligning LLMs with recommendation tasks. [19] studies the binary hypothesis testing for softmax models.

**Reinforcement Learning** In reinforcement learning (RL) [37, 38, 44, 78, 71, 48, 50, 49], an agent learns to make sequential decisions by interacting with an environment to maximize a cumulative reward. RL algorithms, especially policy gradient methods (e.g., REINForCE, PPO, TRPO) [74, 70, 14], often rely on stochastic gradient descent (SGD) or Adam for optimization. Our low-rank optimization techniques for Adam, which could, in theory, be applied to RL training to make policy optimization more memory-efficient.

**Broader Application of Large Language Models** The application of large transformers with long-context capabilities is not limited to language processing; broader domains such as time-series prediction [75, 72, 6, 69] are also of practical importance. Another line of work applies LLM for

structured data understanding [27, 30, 2, 65]. Extending and validating our method to preserve the accuracy of LLMs on these tasks is, therefore, a worthwhile direction for future work.

## D   Limitations

Our work does not have any noteworthy limitations, as we directly address the key limitation of dominant subspace selection in prior work without requiring extra assumptions. In our paper, we provide a theoretical analysis of the convergence rate, though it still relies on the assumptions of $L$-smoothness (Assumption 3.1) and bounded, centered mini-batch gradient noise (Assumption 3.2).

However, we note that these assumptions are standard in the optimization literature.

## E   Societal Impact

Regarding the positive societal impact, by reducing memory requirements for training, our SARA method enables more organizations—including those with limited compute budgets—to train or fine-tune competitive LLMs.

To the best of our knowledge, we do not anticipate any negative societal impacts.

## F   More experimental results

### F.1   High-Rank Updates

Now, we provide more experimental results for high-rank updates.

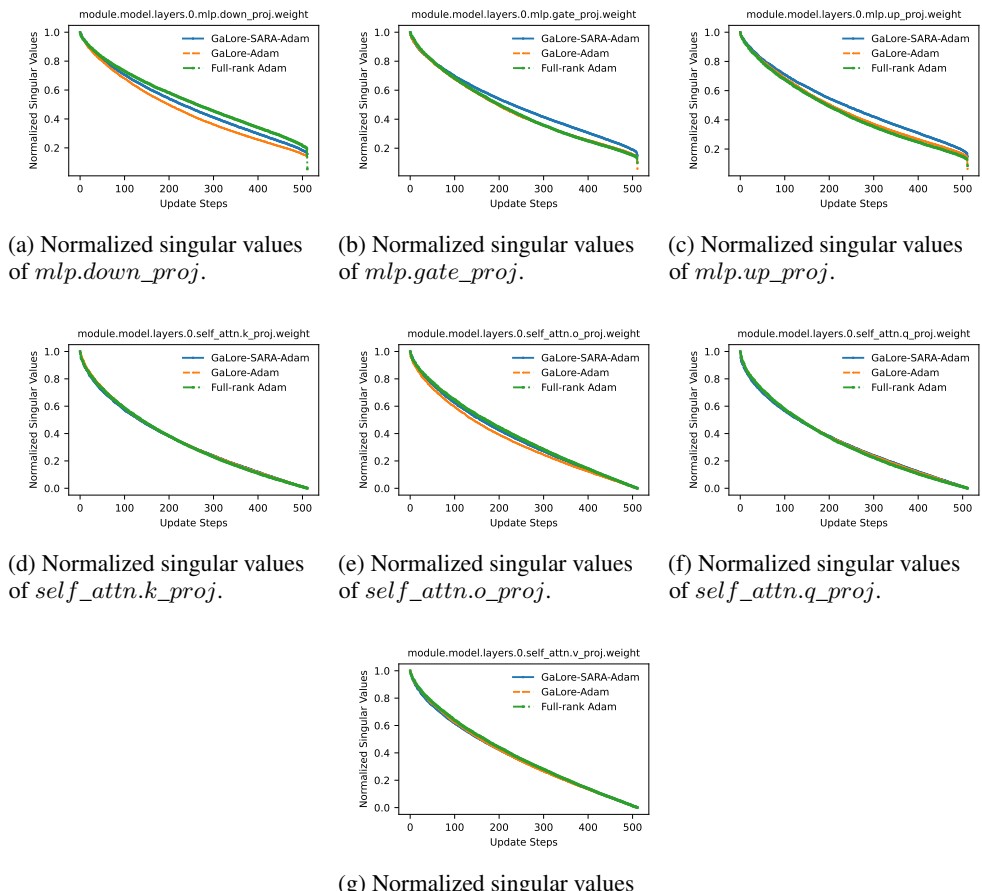

(a) Normalized singular values of $mlp.down\_proj$.

(b) Normalized singular values of $mlp.gate\_proj$.

(c) Normalized singular values of $mlp.up\_proj$.

(d) Normalized singular values of $self\_attn.k\_proj$.

(e) Normalized singular values of $self\_attn.o\_proj$.

(f) Normalized singular values of $self\_attn.q\_proj$.

(g) Normalized singular values of $self\_attn.v\_proj$.

Figure 5: Normalized singular values of the weight difference between the $28k$-step checkpoint and $30k$-step checkpoint in different layers of Block 0 of LLaMA-$60M$ model during pretraining.

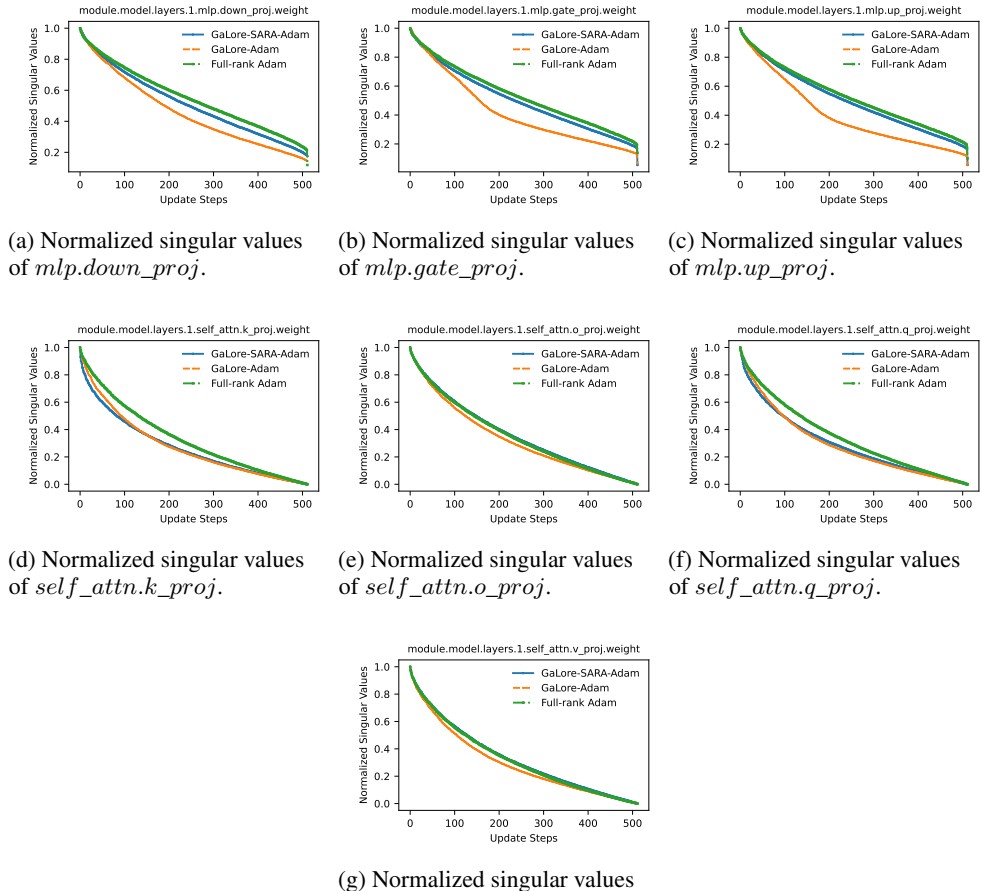

(a) Normalized singular values of $mlp.down\_proj$.

(b) Normalized singular values of $mlp.gate\_proj$.

(c) Normalized singular values of $mlp.up\_proj$.

(d) Normalized singular values of $self\_attn.k\_proj$.

(e) Normalized singular values of $self\_attn.o\_proj$.

(f) Normalized singular values of $self\_attn.q\_proj$.

(g) Normalized singular values of $self\_attn.v\_proj$.

Figure 6: Normalized singular values of the weight difference between the $28k$-step checkpoint and $30k$-step checkpoint in different layers of Block 1 of LLaMA-$60M$ model during pretraining.

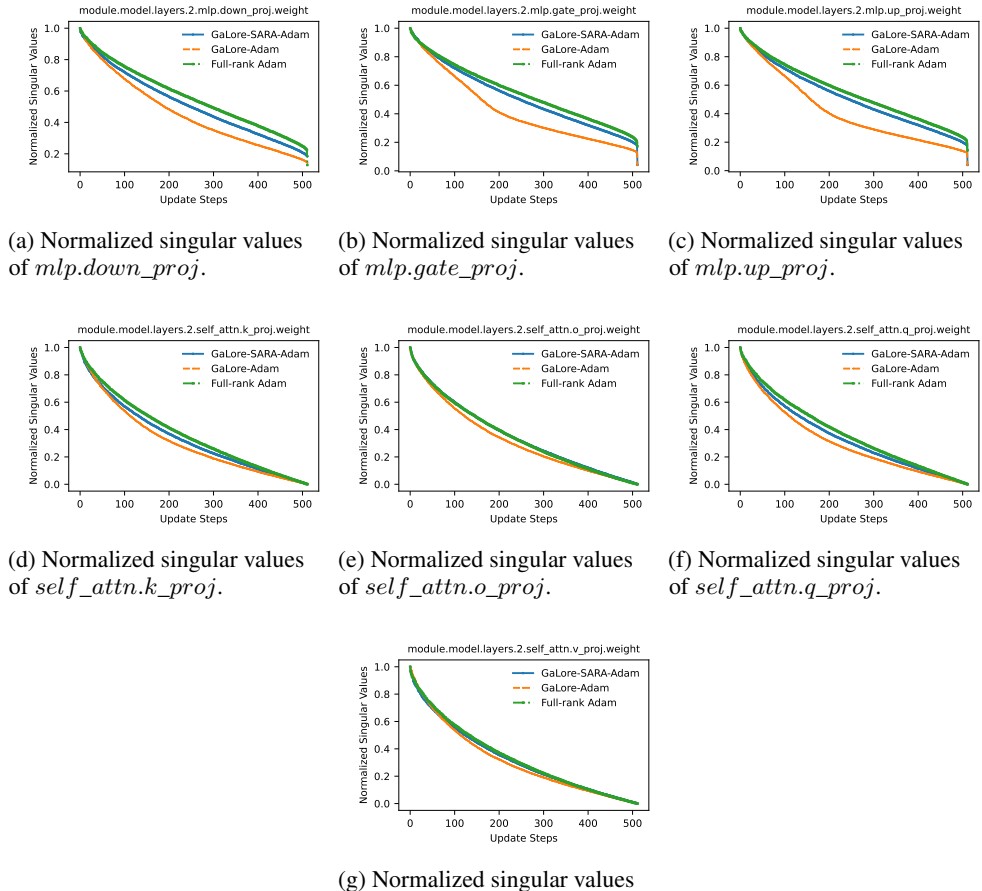

(a) Normalized singular values of $mlp.down\_proj$.

(b) Normalized singular values of $mlp.gate\_proj$.

(c) Normalized singular values of $mlp.up\_proj$.

(d) Normalized singular values of $self\_attn.k\_proj$.

(e) Normalized singular values of $self\_attn.o\_proj$.

(f) Normalized singular values of $self\_attn.q\_proj$.

(g) Normalized singular values of $self\_attn.v\_proj$.

Figure 7: Normalized singular values of the weight difference between the $28k$-step checkpoint and $30k$-step checkpoint in different layers of Block 2 of LLaMA-$60M$ model during pretraining.

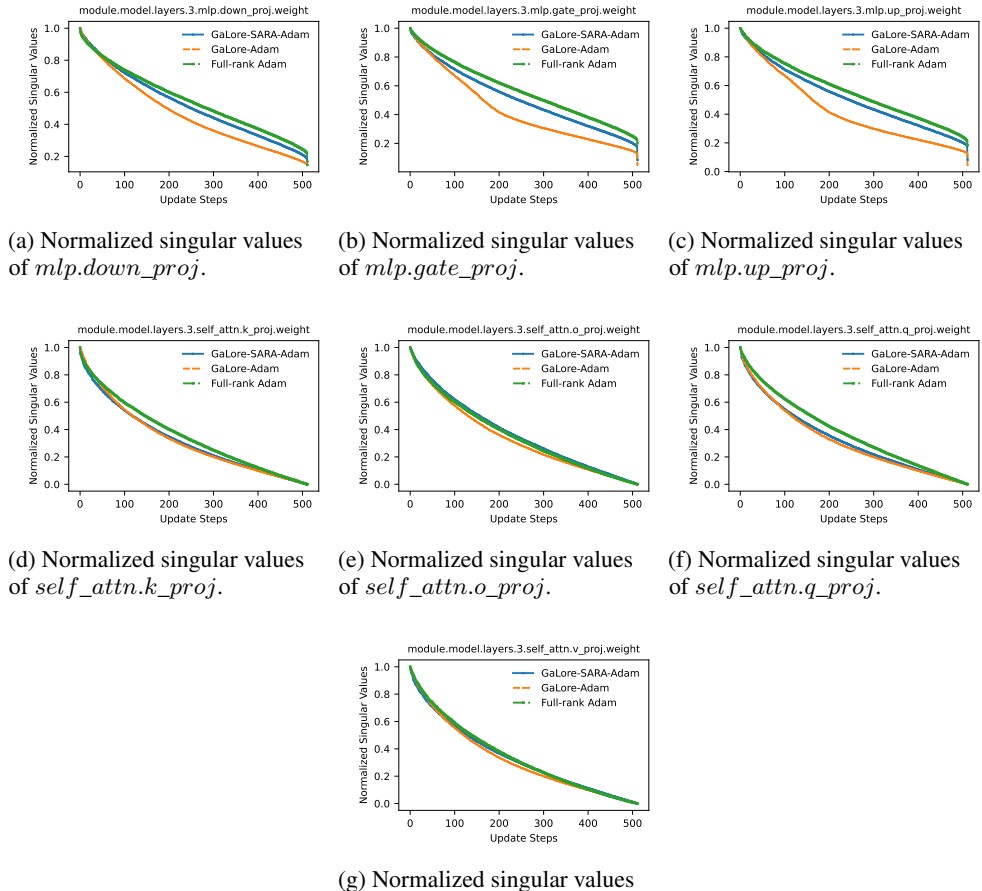

(a) Normalized singular values of $mlp.down\_proj$.

(b) Normalized singular values of $mlp.gate\_proj$.

(c) Normalized singular values of $mlp.up\_proj$.

(d) Normalized singular values of $self\_attn.k\_proj$.

(e) Normalized singular values of $self\_attn.o\_proj$.

(f) Normalized singular values of $self\_attn.q\_proj$.

(g) Normalized singular values of $self\_attn.v\_proj$.

Figure 8: Normalized singular values of the weight difference between the $28k$-step checkpoint and $30k$-step checkpoint in different layers of Block 3 of LLaMA-$60M$ model during pretraining.

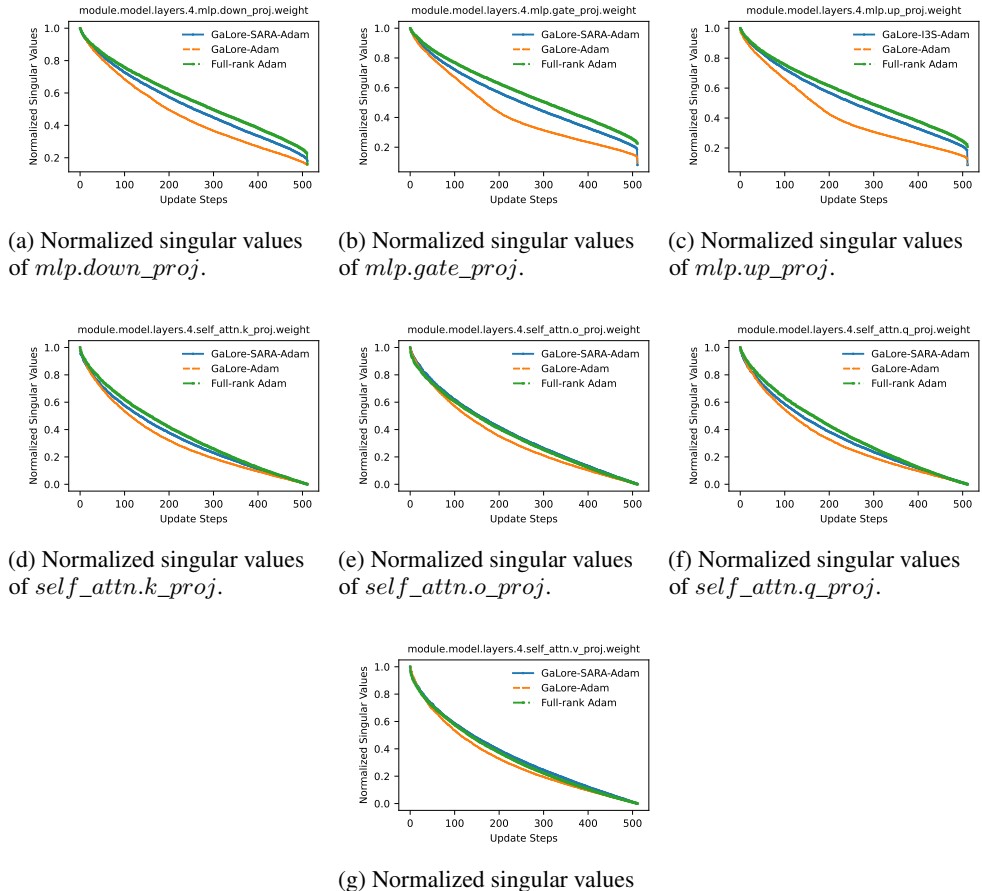

(a) Normalized singular values of $mlp.down\_proj$.

(b) Normalized singular values of $mlp.gate\_proj$.

(c) Normalized singular values of $mlp.up\_proj$.

(d) Normalized singular values of $self\_attn.k\_proj$.

(e) Normalized singular values of $self\_attn.o\_proj$.

(f) Normalized singular values of $self\_attn.q\_proj$.

(g) Normalized singular values of $self\_attn.v\_proj$.

Figure 9: Normalized singular values of the weight difference between the $28k$-step checkpoint and $30k$-step checkpoint in different layers of Block 4 of LLaMA-$60M$ model during pretraining.

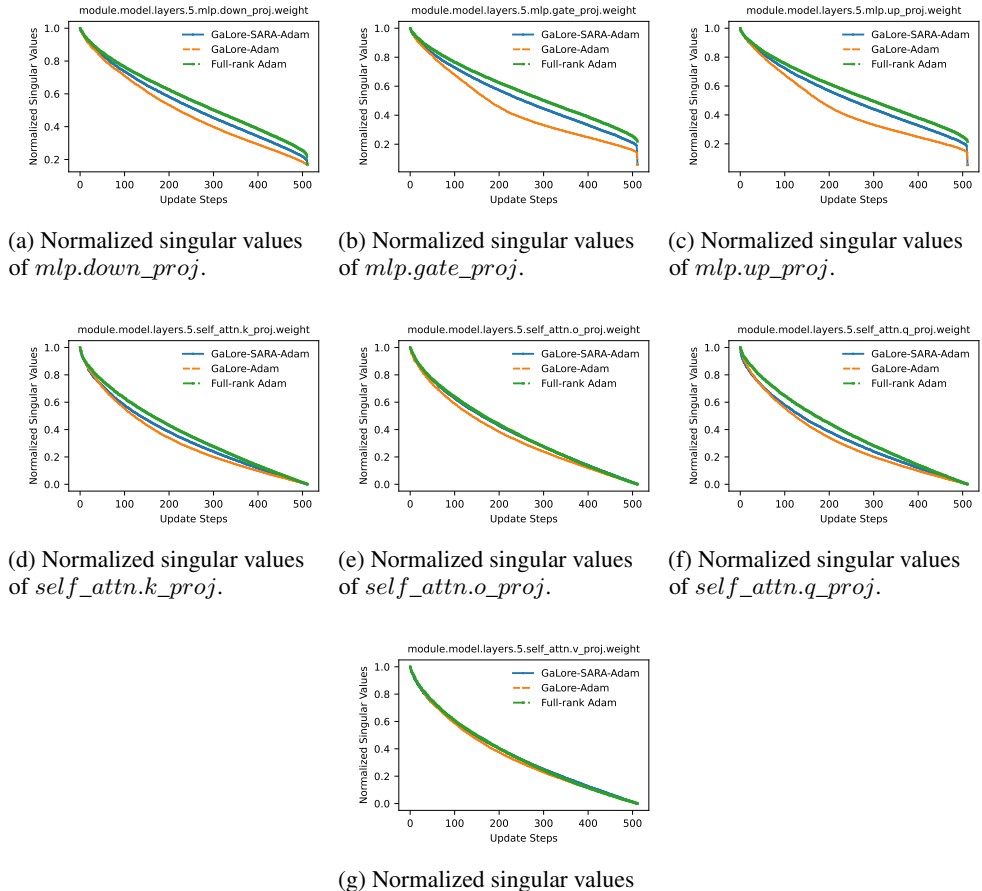

(a) Normalized singular values of $mlp.down\_proj$.

(b) Normalized singular values of $mlp.gate\_proj$.

(c) Normalized singular values of $mlp.up\_proj$.

(d) Normalized singular values of $self\_attn.k\_proj$.

(e) Normalized singular values of $self\_attn.o\_proj$.

(f) Normalized singular values of $self\_attn.q\_proj$.

(g) Normalized singular values of $self\_attn.v\_proj$.

Figure 10: Normalized singular values of the weight difference between the $28k$-step checkpoint and $30k$-step checkpoint in different layers of Block 5 of LLaMA-$60M$ model during pretraining.

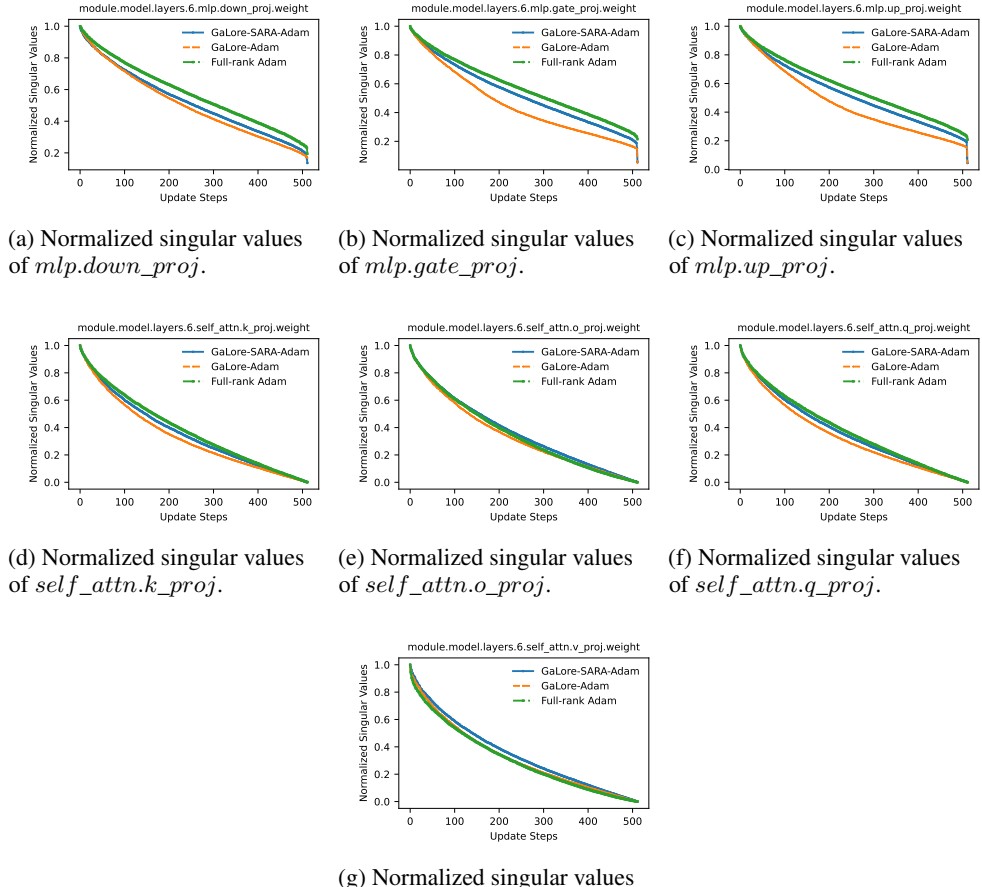

(a) Normalized singular values of $mlp.down\_proj$.

(b) Normalized singular values of $mlp.gate\_proj$.

(c) Normalized singular values of $mlp.up\_proj$.

(d) Normalized singular values of $self\_attn.k\_proj$.

(e) Normalized singular values of $self\_attn.o\_proj$.

(f) Normalized singular values of $self\_attn.q\_proj$.

(g) Normalized singular values of $self\_attn.v\_proj$.

Figure 11: Normalized singular values of the weight difference between the $28k$-step checkpoint and $30k$-step checkpoint in different layers of Block 6 of LLaMA-$60M$ model during pretraining.

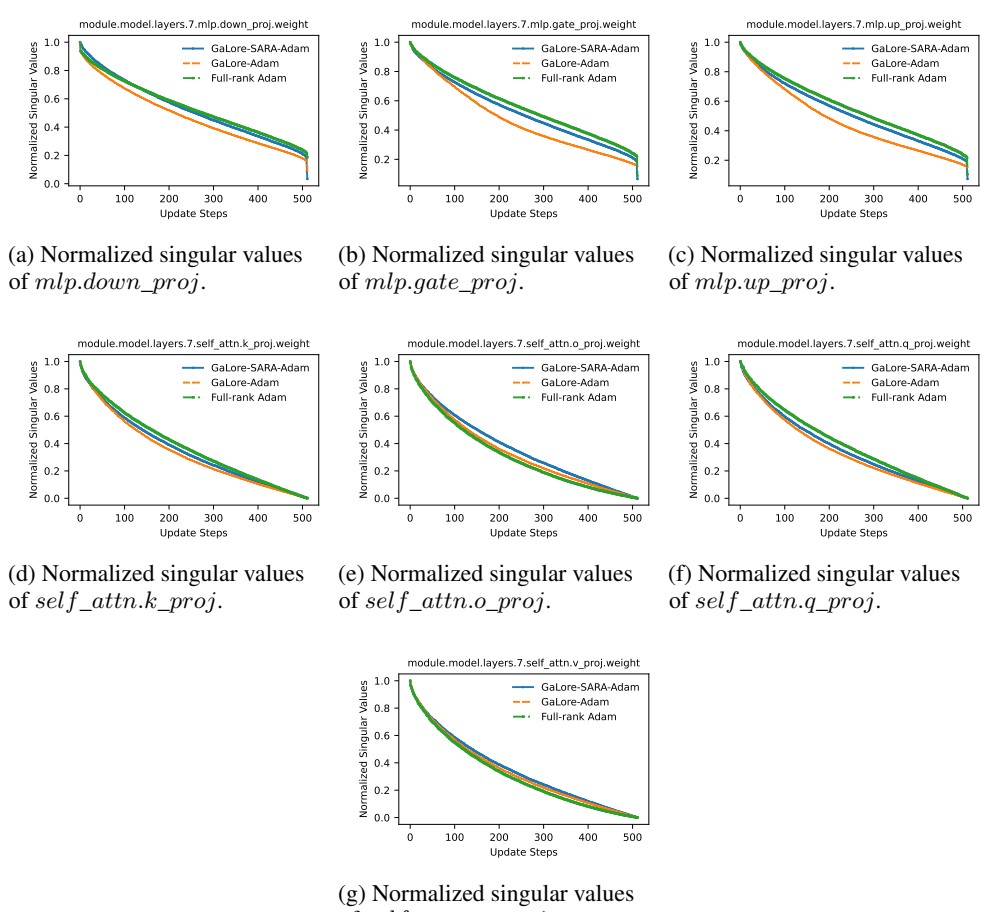

(a) Normalized singular values of $mlp.down\_proj$.

(b) Normalized singular values of $mlp.gate\_proj$.

(c) Normalized singular values of $mlp.up\_proj$.

(d) Normalized singular values of $self\_attn.k\_proj$.

(e) Normalized singular values of $self\_attn.o\_proj$.

(f) Normalized singular values of $self\_attn.q\_proj$.

(g) Normalized singular values of $self\_attn.v\_proj$.

Figure 12: Normalized singular values of the weight difference between the $28k$-step checkpoint and $30k$-step checkpoint in different layers of Block 7 of LLaMA-$60M$ model during pretraining.

## F.2  Anchor Similarity

Now, we provide more experimental results for anchor similarity.

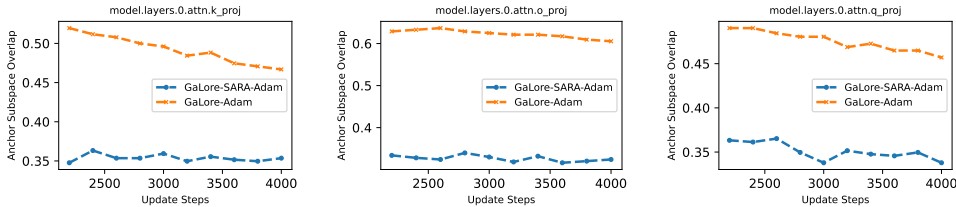

(a) Anchor subspace overlap of *attn.k_proj*.

(b) Anchor subspace overlap of *attn.o_proj*.

(c) Anchor subspace overlap of *attn.q_proj*.

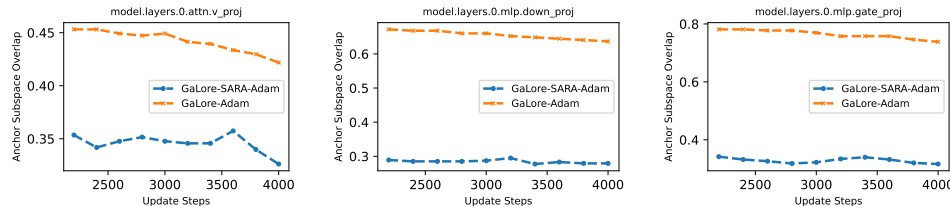

(d) Anchor subspace overlap of *attn.v_proj*.

(e) Anchor subspace overlap of *mlp.down_proj*.

(f) Anchor subspace overlap of *mlp.gate_proj*.

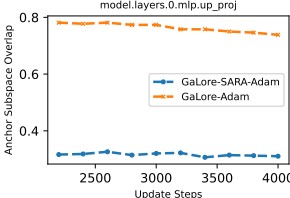

(g) Anchor subspace overlap of *mlp.up_proj*.

Figure 13: The low-rank subspace of different layers in Block 0 at the 2000-th iteration is taken as the anchor subspace. The figure shows the overlap between subspaces of the corresponding layer in Block 0 in later iterations and the anchor subspace.

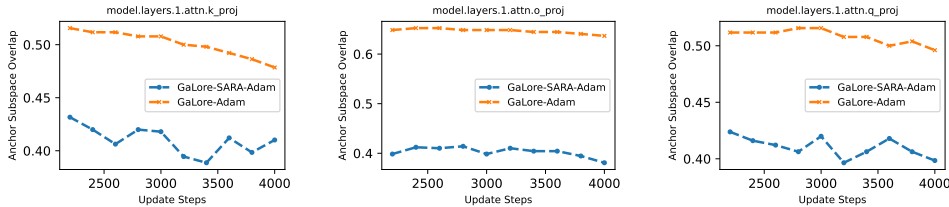

(a) Anchor subspace overlap of *attn.k_proj*.

(b) Anchor subspace overlap of *attn.o_proj*.

(c) Anchor subspace overlap of *attn.q_proj*.

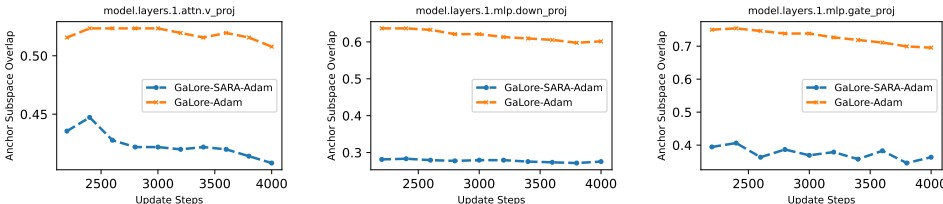

(d) Anchor subspace overlap of *attn.v_proj*.

(e) Anchor subspace overlap of *mlp.down_proj*.

(f) Anchor subspace overlap of *mlp.gate_proj*.

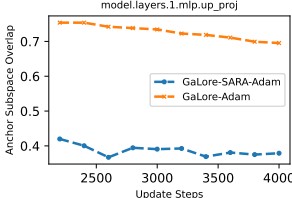

(g) Anchor subspace overlap of *mlp.up_proj*.

Figure 14: The low-rank subspace of different layers in Block 1 at the 2000-th iteration is taken as the anchor subspace. The figure shows the overlap between subspaces of the corresponding layer in Block 1 in later iterations and the anchor subspace.

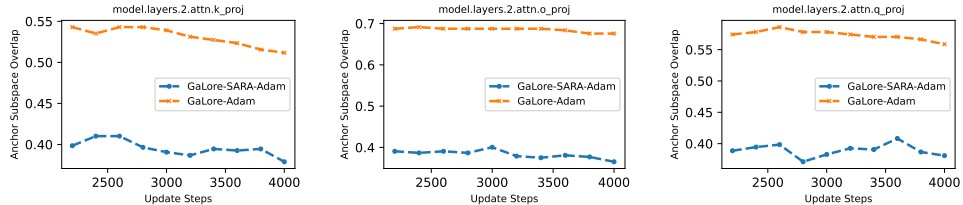

(a) Anchor subspace overlap of *attn.k_proj.*  (b) Anchor subspace overlap of *attn.o_proj.*  (c) Anchor subspace overlap of *attn.q_proj.*

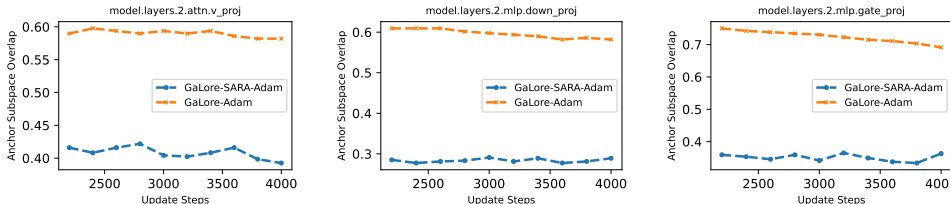

(d) Anchor subspace overlap of *attn.v_proj.*  (e) Anchor subspace overlap of *mlp.down_proj.*  (f) Anchor subspace overlap of *mlp.gate_proj.*

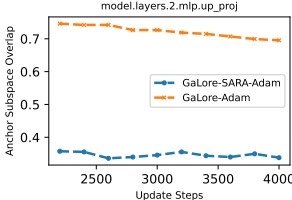

(g) Anchor subspace overlap of *mlp.up_proj.*

Figure 15: The low-rank subspace of different layers in Block 2 at the 2000-th iteration is taken as the anchor subspace. The figure shows the overlap between subspaces of the corresponding layer in Block 2 in later iterations and the anchor subspace.

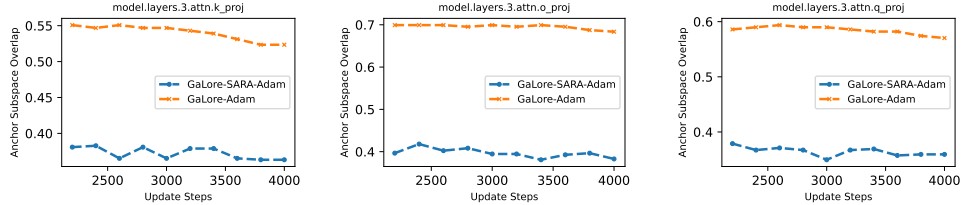

(a) Anchor subspace overlap of *attn.k_proj*.

(b) Anchor subspace overlap of *attn.o_proj*.

(c) Anchor subspace overlap of *attn.q_proj*.

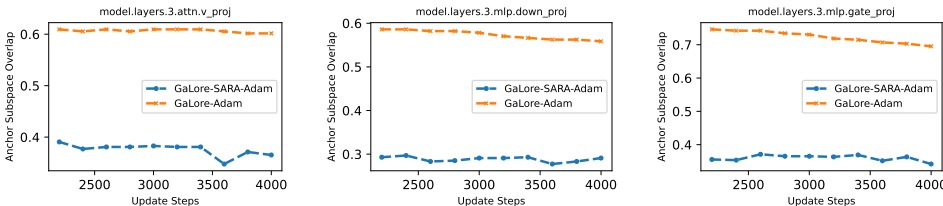

(d) Anchor subspace overlap of *attn.v_proj*.

(e) Anchor subspace overlap of *mlp.down_proj*.

(f) Anchor subspace overlap of *mlp.gate_proj*.

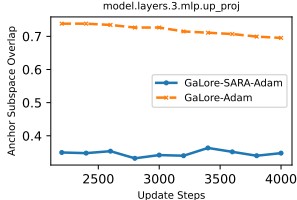

(g) Anchor subspace overlap of *mlp.up_proj*.

Figure 16: The low-rank subspace of different layers in Block 3 at the 2000-th iteration is taken as the anchor subspace. The figure shows the overlap between subspaces of the corresponding layer in Block 3 in later iterations and the anchor subspace.

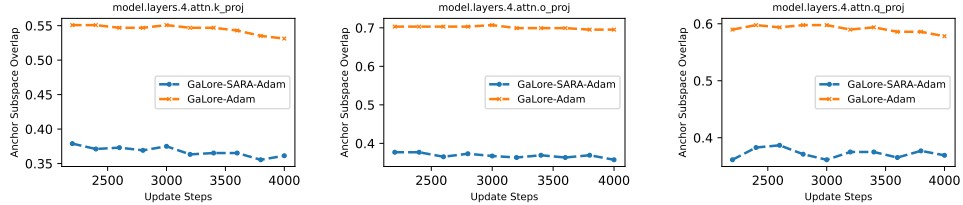

(a) Anchor subspace overlap of *attn.k_proj.*

(b) Anchor subspace overlap of *attn.o_proj.*

(c) Anchor subspace overlap of *attn.q_proj.*

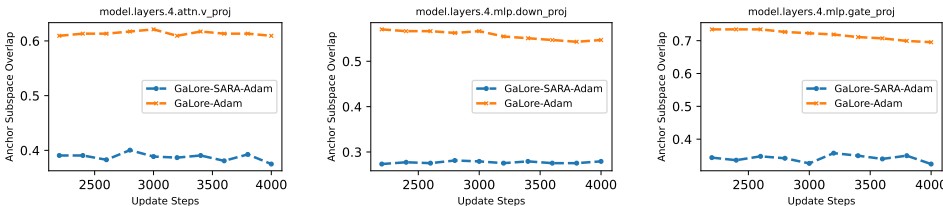

(d) Anchor subspace overlap of *attn.v_proj.*

(e) Anchor subspace overlap of *mlp.down_proj.*

(f) Anchor subspace overlap of *mlp.gate_proj.*

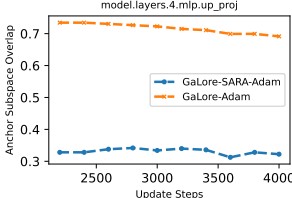

(g) Anchor subspace overlap of *mlp.up_proj.*

Figure 17: The low-rank subspace of different layers in Block 4 at the 2000-th iteration is taken as the anchor subspace. The figure shows the overlap between subspaces of the corresponding layer in Block 4 in later iterations and the anchor subspace.

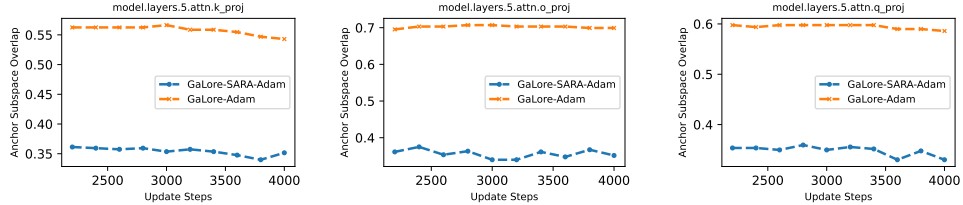

(a) Anchor subspace overlap of *attn.k_proj*.
(b) Anchor subspace overlap of *attn.o_proj*.
(c) Anchor subspace overlap of *attn.q_proj*.

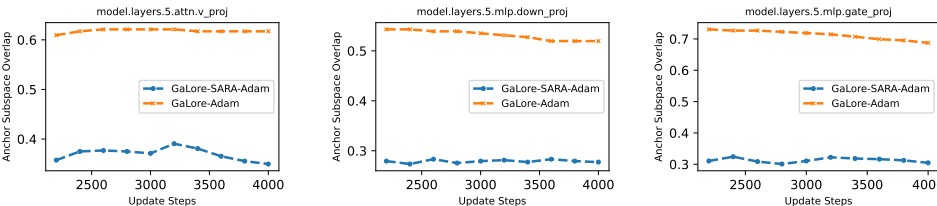

(d) Anchor subspace overlap of *attn.v_proj*.
(e) Anchor subspace overlap of *mlp.down_proj*.
(f) Anchor subspace overlap of *mlp.gate_proj*.

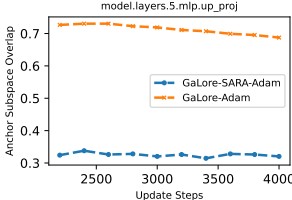

(g) Anchor subspace overlap of *mlp.up_proj*.

Figure 18: The low-rank subspace of different layers in Block 5 at the 2000-th iteration is taken as the anchor subspace. The figure shows the overlap between subspaces of the corresponding layer in Block 5 in later iterations and the anchor subspace.

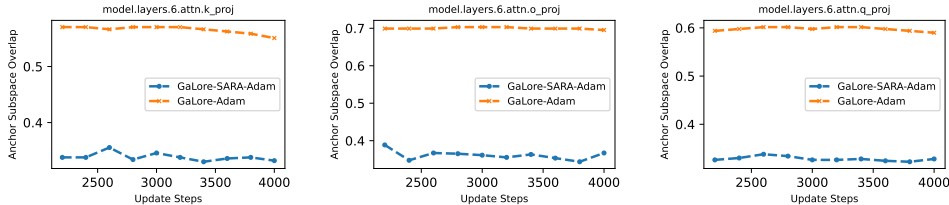

(a) Anchor subspace overlap of *attn.k_proj*.
(b) Anchor subspace overlap of *attn.o_proj*.
(c) Anchor subspace overlap of *attn.q_proj*.

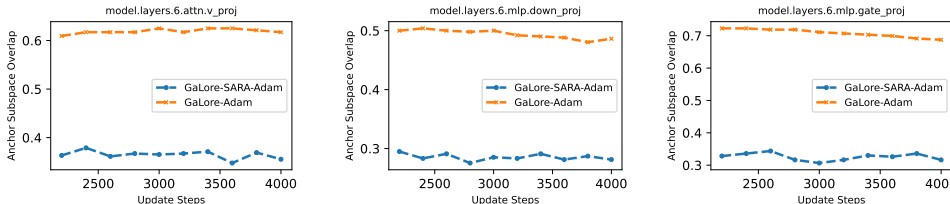

(d) Anchor subspace overlap of *attn.v_proj*.
(e) Anchor subspace overlap of *mlp.down_proj*.
(f) Anchor subspace overlap of *mlp.gate_proj*.

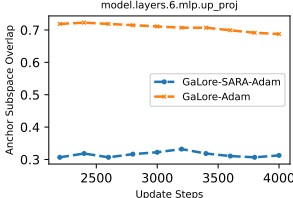

(g) Anchor subspace overlap of *mlp.up_proj*.

Figure 19: The low-rank subspace of different layers in Block 6 at the 2000-th iteration is taken as the anchor subspace. The figure shows the overlap between subspaces of the corresponding layer in Block 6 in later iterations and the anchor subspace.

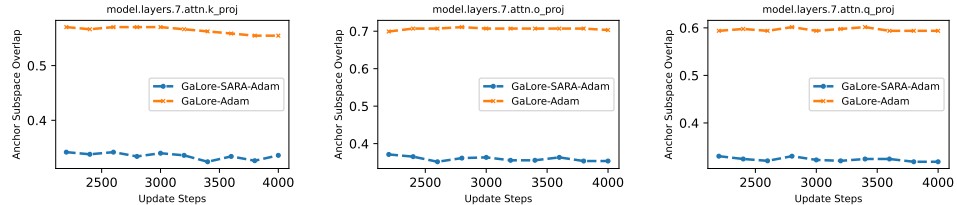

(a) Anchor subspace overlap of *attn.k_proj.*
(b) Anchor subspace overlap of *attn.o_proj.*
(c) Anchor subspace overlap of *attn.q_proj.*

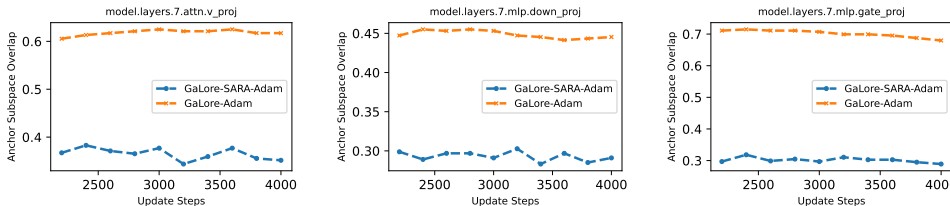

(d) Anchor subspace overlap of *attn.v_proj.*
(e) Anchor subspace overlap of *mlp.down_proj.*
(f) Anchor subspace overlap of *mlp.gate_proj.*

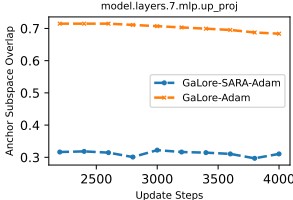

(g) Anchor subspace overlap of *mlp.up_proj.*

Figure 20: The low-rank subspace of different layers in Block 7 at the 2000-th iteration is taken as the anchor subspace. The figure shows the overlap between subspaces of the corresponding layer in Block 7 in later iterations and the anchor subspace.

## F.3    Adjacent Overlap

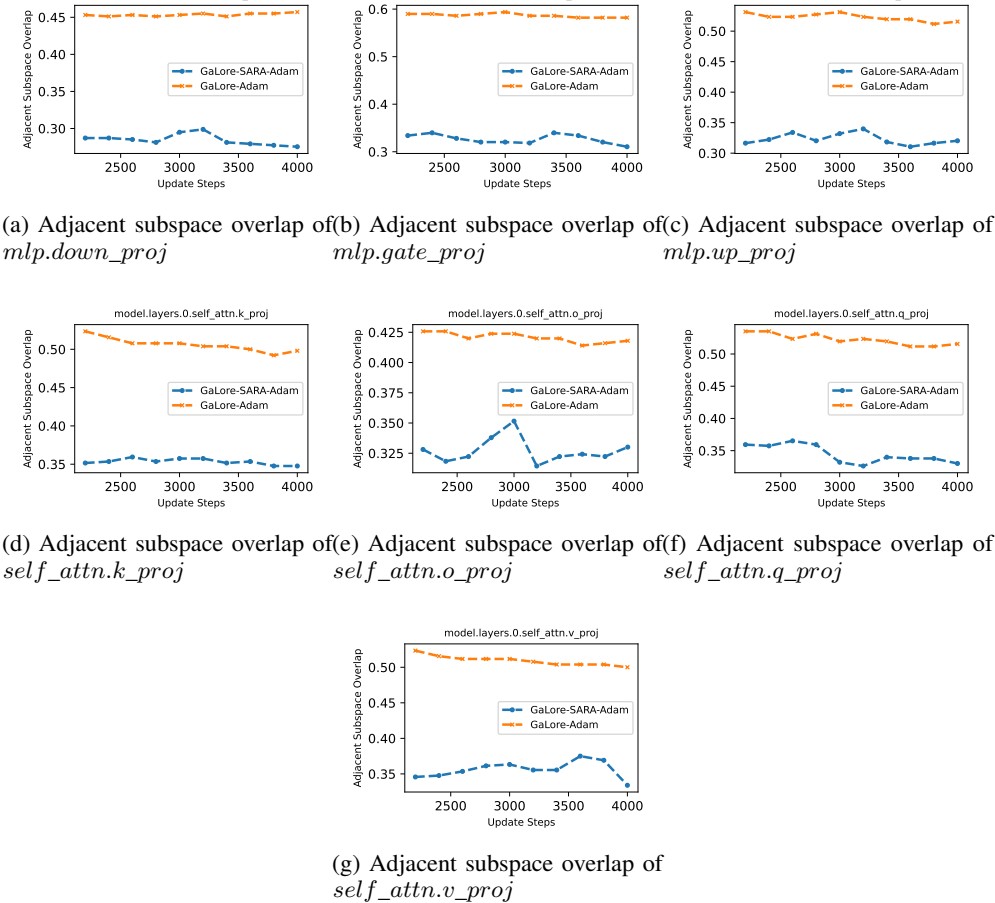

(a) Adjacent subspace overlap of $mlp.down\_proj$

(b) Adjacent subspace overlap of $mlp.gate\_proj$

(c) Adjacent subspace overlap of $mlp.up\_proj$

(d) Adjacent subspace overlap of $self\_attn.k\_proj$

(e) Adjacent subspace overlap of $self\_attn.o\_proj$

(f) Adjacent subspace overlap of $self\_attn.q\_proj$

(g) Adjacent subspace overlap of $self\_attn.v\_proj$

Figure 21: The overlap between adjacent subspaces of optimization trajectory of different layers in Block 0 in GaLore-Adam and GaLore-SARA-Adam during pretraining on the LLaMA-60M model between 2200-th and 4000-th iteration

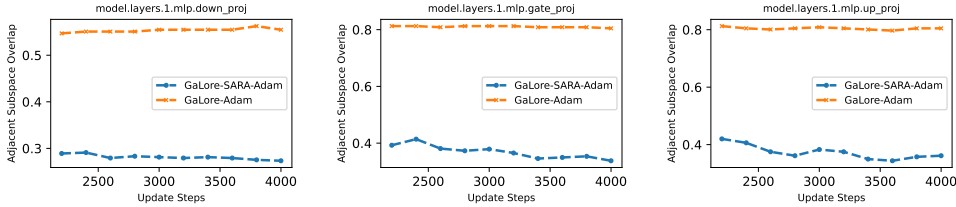

(a) Adjacent subspace overlap of $mlp.down\_proj$ (b) Adjacent subspace overlap of $mlp.gate\_proj$ (c) Adjacent subspace overlap of $mlp.up\_proj$

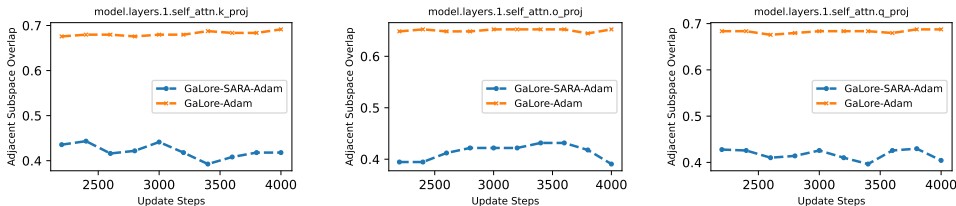

(d) Adjacent subspace overlap of $self\_attn.k\_proj$ (e) Adjacent subspace overlap of $self\_attn.o\_proj$ (f) Adjacent subspace overlap of $self\_attn.q\_proj$

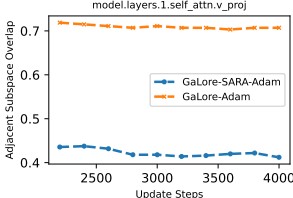

(g) Adjacent subspace overlap of $self\_attn.v\_proj$

Figure 22: The overlap between adjacent subspaces of optimization trajectory of different layers in Block 1 in GaLore-Adam and GaLore-SARA-Adam during pretraining on the LLaMA-60M model between 2200-th and 4000-th iteration

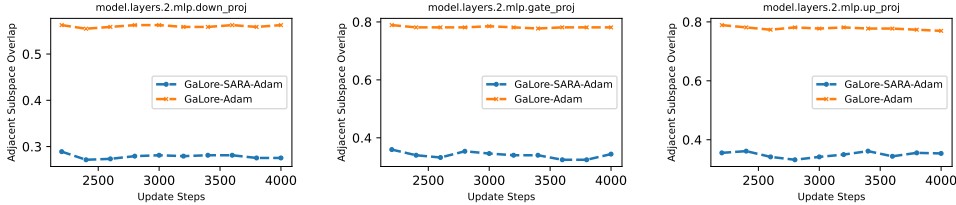

(a) Adjacent subspace overlap of $mlp.down\_proj$

(b) Adjacent subspace overlap of $mlp.gate\_proj$

(c) Adjacent subspace overlap of $mlp.up\_proj$

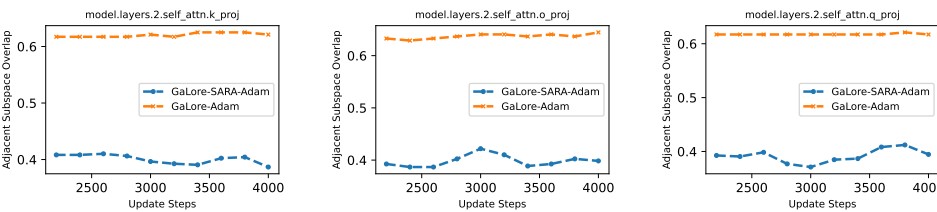

(d) Adjacent subspace overlap of $self\_attn.k\_proj$

(e) Adjacent subspace overlap of $self\_attn.o\_proj$

(f) Adjacent subspace overlap of $self\_attn.q\_proj$

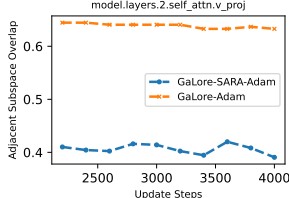

(g) Adjacent subspace overlap of $self\_attn.v\_proj$

Figure 23: The overlap between adjacent subspaces of optimization trajectory of different layers in Block 2 in GaLore-Adam and GaLore-SARA-Adam during pretraining on the LLaMA-60M model between 2200-th and 4000-th iteration

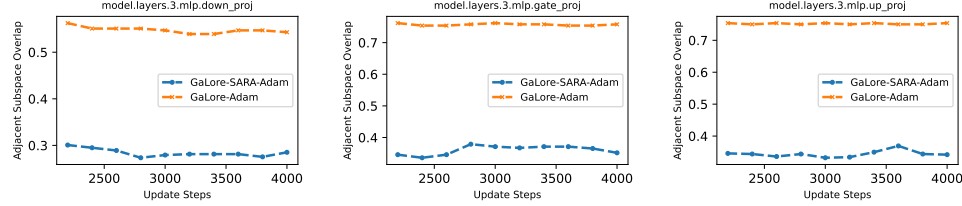

(a) Adjacent subspace overlap of *mlp.down_proj* (b) Adjacent subspace overlap of *mlp.gate_proj* (c) Adjacent subspace overlap of *mlp.up_proj*

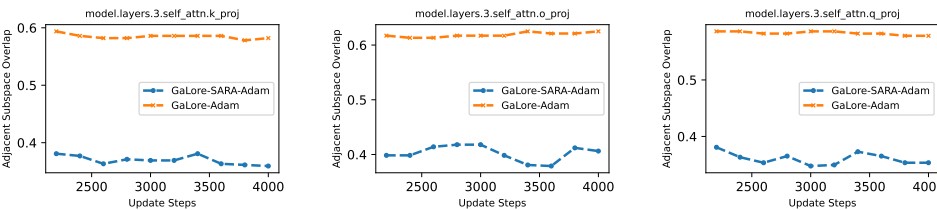

(d) Adjacent subspace overlap of *self_attn.k_proj* (e) Adjacent subspace overlap of *self_attn.o_proj* (f) Adjacent subspace overlap of *self_attn.q_proj*

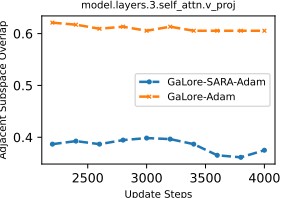

(g) Adjacent subspace overlap of *self_attn.v_proj*

Figure 24: The overlap between adjacent subspaces of optimization trajectory of different layers in Block 3 in GaLore-Adam and GaLore-SARA-Adam during pretraining on the LLaMA-60M model between 2200-th and 4000-th iteration

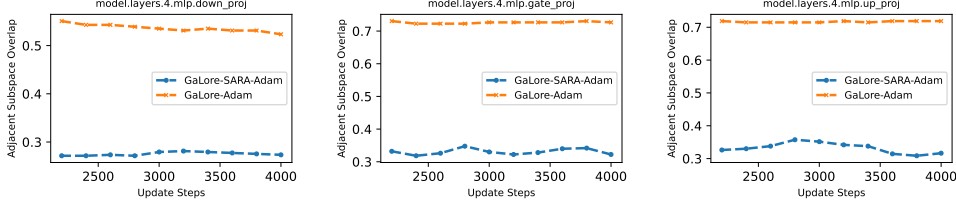

(a) Adjacent subspace overlap of $mlp.down\_proj$ (b) Adjacent subspace overlap of $mlp.gate\_proj$ (c) Adjacent subspace overlap of $mlp.up\_proj$

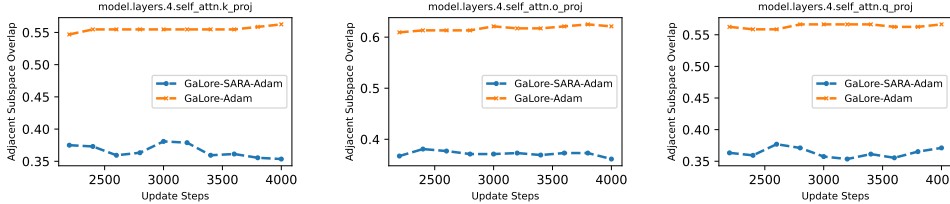

(d) Adjacent subspace overlap of $self\_attn.k\_proj$ (e) Adjacent subspace overlap of $self\_attn.o\_proj$ (f) Adjacent subspace overlap of $self\_attn.q\_proj$

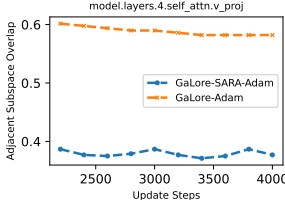

(g) Adjacent subspace overlap of $self\_attn.v\_proj$

Figure 25: The overlap between adjacent subspaces of optimization trajectory of different layers in Block 4 in GaLore-Adam and GaLore-SARA-Adam during pretraining on the LLaMA-60M model between 2200-th and 4000-th iteration

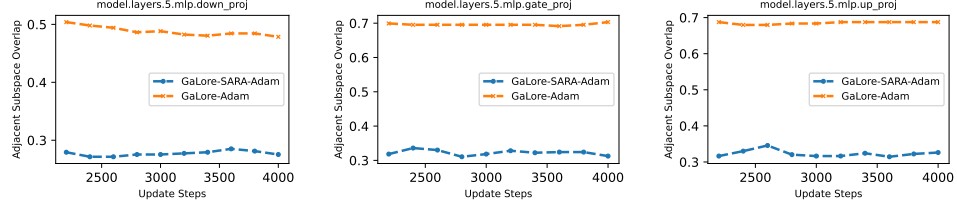

(a) Adjacent subspace overlap of $mlp.down\_proj$

(b) Adjacent subspace overlap of $mlp.gate\_proj$

(c) Adjacent subspace overlap of $mlp.up\_proj$

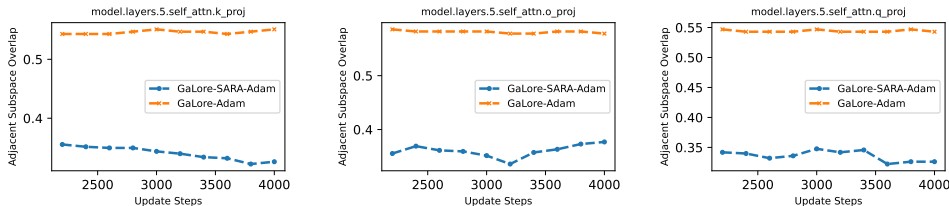

(d) Adjacent subspace overlap of $self\_attn.k\_proj$

(e) Adjacent subspace overlap of $self\_attn.o\_proj$

(f) Adjacent subspace overlap of $self\_attn.q\_proj$

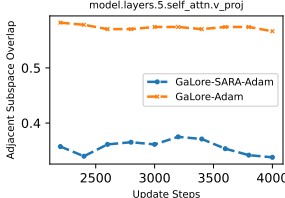

(g) Adjacent subspace overlap of $self\_attn.v\_proj$

Figure 26: The overlap between adjacent subspaces of optimization trajectory of different layers in Block 5 in GaLore-Adam and GaLore-SARA-Adam during pretraining on the LLaMA-60M model between 2200-th and 4000-th iteration

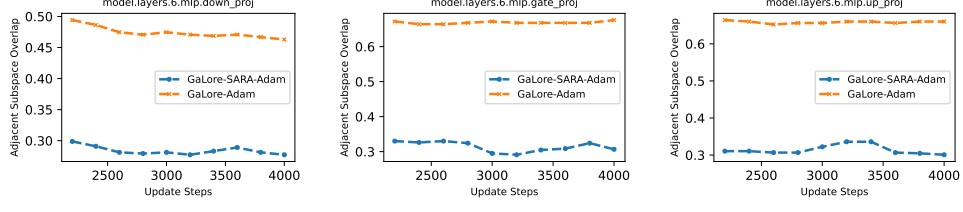

(a) Adjacent subspace overlap of $mlp.down\_proj$ (b) Adjacent subspace overlap of $mlp.gate\_proj$ (c) Adjacent subspace overlap of $mlp.up\_proj$

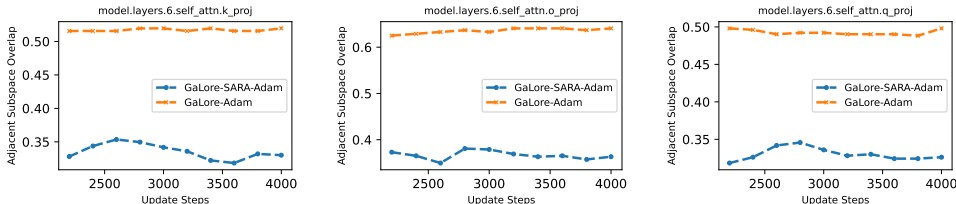

(d) Adjacent subspace overlap of $self\_attn.k\_proj$ (e) Adjacent subspace overlap of $self\_attn.o\_proj$ (f) Adjacent subspace overlap of $self\_attn.q\_proj$

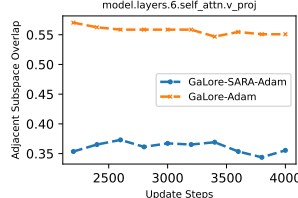

(g) Adjacent subspace overlap of $self\_attn.v\_proj$

Figure 27: The overlap between adjacent subspaces of optimization trajectory of different layers in Block 6 in GaLore-Adam and GaLore-SARA-Adam during pretraining on the LLaMA-60M model between 2200-th and 4000-th iteration

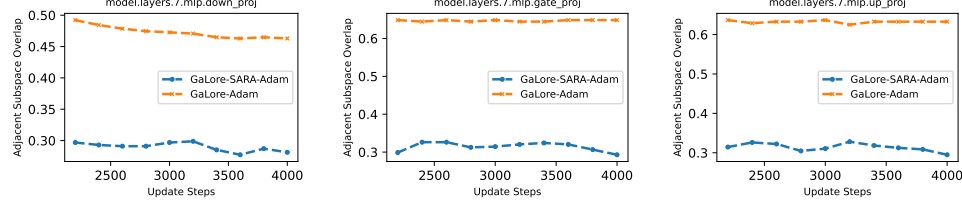

(a) Adjacent subspace overlap of (b) Adjacent subspace overlap of (c) Adjacent subspace overlap of
$mlp.down\_proj$          $mlp.gate\_proj$          $mlp.up\_proj$

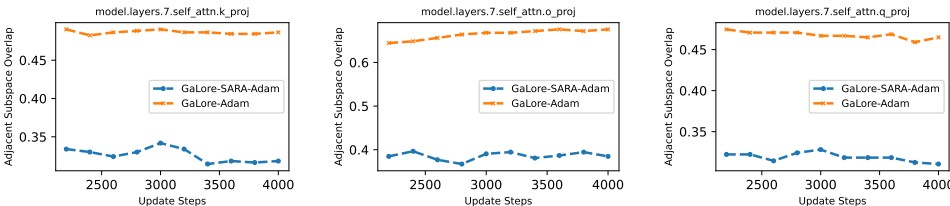

(d) Adjacent subspace overlap of (e) Adjacent subspace overlap of (f) Adjacent subspace overlap of
$self\_attn.k\_proj$        $self\_attn.o\_proj$        $self\_attn.q\_proj$

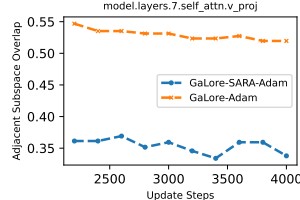

(g) Adjacent subspace overlap of
$self\_attn.v\_proj$

Figure 28: The overlap between adjacent subspaces of optimization trajectory of different layers in Block 7 in GaLore-Adam and GaLore-SARA-Adam during pretraining on the LLaMA-60M model between 2200-th and 4000-th iteration

