# OpenReview forum: "Breaking the Frozen Subspace: Importance Sampling for Low-Rank Optimization in LLM Pretraining"
_NeurIPS.cc/2025/Conference — NeurIPS 2025 poster_

### Official Review · Reviewer_TBg6 · 2025-06-27

**Clarity:** 3
**Significance:** 2
**Originality:** 3
**Rating:** 3
**Confidence:** 3

**Summary:**

This paper identifies and addresses the "frozen subspace" problem in low-rank optimization methods used for LLM pretraining. The authors observe that methods like GaLore, which rely on projecting gradients onto a dominant subspace, often suffer from this subspace becoming static early in training, thus limiting the model's learning capacity. To counteract this, the paper proposes SARA (Importance Sampling for Low-Rank optimization), which replaces the deterministic selection of top singular vectors with a stochastic, importance-sampling-based approach. By sampling basis vectors with probabilities proportional to their singular values, SARA encourages the exploration of more diverse subspaces throughout training. The work is supported by a theoretical convergence guarantee and empirical results on LLaMA models, demonstrating that SARA consistently improves performance over dominant subspace baselines and narrows the gap with full-rank Adam.

**Questions:**

Please refer to the Weaknesses.

**Ethical Concerns:**

["NO or VERY MINOR ethics concerns only"]

**Final Justification:**

I keep my score unchanged. The authors have addressed some of my concerns, but others remain. E.g., the experimental setting is not reasonable, and I have concerns on the scalability of the method. It is crucial to test models on a large model, at least 7B.

**Limitations:**

Yes

**Quality:**

2

**Strengths And Weaknesses:**

This paper presents a contribution to memory-efficient optimization, with several key strengths but also noteworthy weaknesses that temper its overall impact.

**Strengths**

1. Clear Motivation and Problem Diagnosis: The identification and empirical illustration of the "frozen dominant subspace" phenomenon provide a compelling rationale for the proposed work.

2. Simple and Practical Method: As a plug-and-play module that modifies only the subspace selection step, SARA is highly practical and can be easily integrated into existing low-rank optimization frameworks like GaLore and Fira.

3. Theoretical Backing: Providing a formal convergence guarantee (Theorem 3.4) is a significant strength. It adds a layer of theoretical rigor that is absent in the popular dominant subspace approach, bolstering the credibility of the method.

4. Good Clarity and Presentation: The paper is well-written and easy to follow. The problem, method, and results are presented logically and supported by clear figures and algorithms.

**Weaknesses**

1. Incremental Novelty: The core idea of analyzing and modifying low-rank subspaces to improve performance is an established paradigm, particularly in work on improving LoRA. This positions the contribution as an effective but arguably incremental advancement rather than a fundamentally new approach.

2. Limited Experimental Scope and Scale: The empirical evaluation, while covering multiple optimizer variants, lacks the scale and breadth expected of top-tier work in this area.

i) The experiments do not extend to larger, more commonly benchmarked models like a LLaMA-7B, and larger datasets, which would be a more convincing test of scalability.

ii) The evaluation is confined to perplexity. It critically lacks evaluation on a suite of downstream fine-tuning tasks (e.g. Avg Acc), which is the standard for demonstrating the quality and generalizability of a pretrained model.

iii) There is a crucial absence of any ablation or sensitivity analysis on the rank r. (a key hyperparameter)

3. Marginal Performance Gains: The practical significance of the improvements is questionable in some cases. For instance, on the 130M model, SARA's perplexity gain over the Fira-Adam baseline is minimal (22.22 vs. 22.37). Maybe the SOTA baseline has implicitly solved the subspace problem.

4. "No Overhead" Claim: The paper claims SARA "does not bring extra overhead". However, as the method introduces an explicit weighted sampling step after the SVD. This claim requires empirical validation with wall-clock time and memory usage comparisons to prove that the additional computational cost is indeed negligible.

5. The paper's own theory suggests SARA's convergence rate is slightly worse than GoLore's by a constant factor , yet SARA consistently outperforms GoLore in experiments. This gap is not adequately discussed, leaving open questions about the factors driving the empirical success.

---

> ### Author Rebuttal · Authors · 2025-07-29
>
> We express our deepest gratitude to the reviewer for the time and effort in reviewing our work. It is very encouraging for us to see your evaluation of our work as 1). having clear motivation, 2). simple and practical, 3). theoretically rigorous, and 4). having clear presentation. Below, we want to address your concern about our work.
>
> **Weakness 1**: The idea of analyzing low-rank subspaces to improve performance is an established paradigm, particularly in work on improving LoRA. This positions the contribution as an effective but arguably incremental advancement.
>
> **Response to Weakness 1**: We appreciate your observation that analyzing and modifying low-rank subspaces is an established paradigm, especially in the context of LoRA. We agree that this is a widely explored area. However, we would like to clarify that the main novelty of our work is to address a newly identified challenge within this paradigm—namely, the frozen dominant subspace phenomenon.
>
> This phenomenon was recently formalized in [1, 2], and to the best of our knowledge, our paper is the first to propose a theoretically grounded and empirically validated method to mitigate it. Specifically, while prior methods like GaLore and $Q$-GaLore project gradients onto the dominant subspace (defined by leading singular vectors), we show empirically (e.g., Figure 3, Table 1) that such a strategy can lead to stagnation in subspace evolution during pretraining, ultimately limiting the expressiveness of weight updates.
>
> To tackle this, we introduce an importance sampling mechanism inspired by classical sketching techniques in randomized linear algebra. Given a big data matrix $A \in \mathbb{R}^{n \times d}$, sketching aims to reduce dimensionality by sampling a small matrix $SA$, where $S \in \mathbb{R}^{s \times n}$ and $s \ll n$. A principled way to construct $S$ is by leveraging leverage scores, which quantify each row’s contribution to the row space of $A$. Rather than selecting the top-$k$ leverage rows (which could lead to overfitting to a narrow subspace), sketching theory recommends sampling **proportionally** to leverage scores, which better preserves the full subspace structure [3].
>
> We draw a direct analogy between this and our method: we treat the singular values (quantifying each singular vector’s contribution) of the gradient as analogous to leverage scores, and sample left singular vectors without replacement, **proportional** to their singular values. This avoids always selecting the same top-$k$ directions, like in GaLore and $Q$-GaLore, and instead promotes diversity across training steps. Our importance sampling method is not a heuristic adjustment made purely for empirical performance. Rather, it is a theoretically motivated approach rooted in randomized linear algebra and directly addresses the frozen subspace problem. While LoRA-related work focuses on augmenting low-rank structure, our work focuses on dynamically diversifying it during optimization, which is a crucial distinction. As far as we know, no prior work explores importance-sampled low-rank subspace selection in this way, particularly within the context of LLM pretraining.
>
> Moreover, SVD is a well-established and widely used technique in machine learning. Many methods—such as principal component analysis (PCA) [4] and attention approximation techniques [5]—rely on projecting data onto the top-$k$ left singular vectors of a data matrix to reduce dimensionality. Our randomized importance sampling strategy offers a novel perspective: instead of deterministically selecting the top directions, probabilistic selection may enhance subspace diversity and performance. We believe this insight could inspire future work in improving other applications that currently rely on fixed top-$k$ subspace projections and in theoretically and/or empirically studying the similarities and differences between our importance sampling and leverage score sampling.
>
> **Weakness 2**:
>
> **2a**. The experiments do not extend to larger models.
>
> **2b**. The evaluation is confined to perplexity. It critically lacks evaluation on a suite of downstream fine-tuning tasks, which is the standard for demonstrating the quality and generalizability of a pretrained model.
>
> **2c**. There is a crucial absence of any ablation or sensitivity analysis on the rank $r$. (a key hyperparameter).
>
> **Response to Weakness 2**: Thank you very much for your insightful comments. We currently report experimental results on models with 60M, 130M, 350M, and 1.1B parameters (Tables 1 and 2), demonstrating that SARA scales effectively across a range of model sizes. Due to limitations in time and computational resources, it is very challenging to present results on larger models at this stage. Even conducting experiments on the 1.1B model takes more than 10 days by using all of our GPUs. Nevertheless, we would like to emphasize that our theoretical convergence guarantee (Theorem 3.4) is independent of model size and thus supports general scalability.
>
> Although we lack the time and resources to run very large-scale experiments, we have fully utilized the rebuttal period to present the following experimental results that are computationally efficient and feasible within this limited timeframe. The following results can address your concerns in **2b** and **2c**.
>
> $\bullet$ **Hyperparameter sensitivity**: Demonstrated stable performance across critical hyperparameters (see our **Response to Weakness and Question 2** to **Reviewer ZP7Q**).
>
> $\bullet$ **Additional metrics beyond validation perplexity**: Confirmed enhanced downstream task performance (see our **Response to Weakness and Question 1** to **Reviewer ZP7Q**).
>
> $\bullet$ **Comparison with uniform sampling**: Highlighted advantages of our sampling strategy (see our **Response to Weakness and Question 2** to **Reviewer yFMY**).
>
> $\bullet$ **Multiple random seeds**: Verified the robustness and reliability of our method across different runs (see our **Response to Weakness and Question 2** to **Reviewer yFMY**).
>
> **Weakness 3**: The practical significance of the improvements is questionable in some cases. On the 130M model, SARA's perplexity gain over the Fira-Adam baseline is minimal (22.22 vs. 22.37).
>
> **Response to Weakness 3**: We thank you very much for pointing this out. We want to highlight that the key measure is not the raw perplexity improvement, but the reduction in the performance gap between the low-rank optimizer and full-rank Adam. In the 130M case, SARA reduces this gap by 42.25\%, which is substantial given the tight margins typically observed in large-scale LLM pretraining. Also, SARA consistently improves performance across a wide range of model sizes (60M to 1.1B) and optimizer variants (Fira, GaLore, Adafactor, Adam-mini, 8-bit), as shown in Table 1 and Table 2. This consistency indicates that the improvements are not isolated noise but the result of meaningful enhancements to subspace exploration during optimization.
>
> Additionally, we only pretrain LLaMA models with a certain number of tokens, so we can only observe relatively small performance gains in our paper. In fact, the performance gain of our method compared to GaLore increases as the language models are trained with more tokens. Therefore, once the language models are fully trained on the pretraining task, we expect to see much larger performance gains.
>
> **Weakness 4**: "The paper claims SARA "does not bring extra overhead". This claim requires empirical validation with wall-clock time and memory usage comparisons.
>
>
> **Response to Weakness 4**: Thank you for pointing this out. Therefore, we conducted an additional experiment to address your concern. In the experiment, we measure the wall-clock time of SVD computation and sampling. We calculate the average wall-clock time of ten runs; the result for SVD is 0.34 seconds for a 2048 by 2048 matrix, and the result for sampling is 0.0005 seconds. The empirical result shows that the additional computational cost brought by sampling is negligible.
>
> **Weakness 5**: The paper's own theory suggests SARA's convergence rate is slightly worse than GoLore's by a constant factor, yet SARA consistently outperforms GoLore in experiments.
>
> **Response to Weakness 5**: We thank you very much for pointing out this important point. SARA picks singular vectors proportional to their singular values: this can not only ensure that we pick useful information (high singular values) but also address frozen dominant subspace phenomenon. GoLore, on the other hand, projects gradients onto a random low-rank subspace using unbiased random projections. This random projection cannot guarantee that they can pick so useful information compared to our SARA. Additionally, our theoretical guarantee considers the worst case. In practice, "the gradient lies
> predominantly in the smaller, top subspace" [2], and our importance sampling concentrates on informative directions, leading to a better effective convergence behavior than the worst-case suggests. We have included the above discussion to our paper.
>
> Once again, we are deeply grateful for your insightful comments. We have addressed all identified weaknesses and questions, and we respectfully and genuinely hope you will consider raising your score.
>
> [1] Zhenyu Zhang, Ajay Jaiswal, Lu Yin, Shiwei Liu, Jiawei Zhao, Yuandong Tian, and Zhangyang Wang. "Q-galore: Quantized galore with int4 projection and layer-adaptive low-rank gradients." Preprint'24.
>
> [2] Guy Gur-Ari, Daniel A. Roberts, and Ethan Dyer. "Gradient descent happens in a tiny subspace." Preprint'18.
>
> [3] Michael W. Mahoney "Randomized algorithms for matrices and data." Foundations and Trends in Machine Learning'11.
>
> [4] Jianqing Fan, Qiang Sun, Wen-Xin Zhou, and Ziwei Zhu. "Principal component analysis for big data." Preprint'18.
>
> [5] Sinong Wang, Belinda Z. Li, Madian Khabsa, Han Fang, and Hao Ma. "Linformer: Self-attention with linear complexity." Preprint'20.

---

> > ### Comment · Reviewer_TBg6 · 2025-08-05
> >
> > Thank you for the response. I will keep my score unchanged.

---

> > > ### Author Response · Authors · 2025-08-05
> > > **Thank you very much!**
> > >
> > > We sincerely thank you for your time in reading our response.

---

### Official Review · Reviewer_yFMY · 2025-06-30

**Clarity:** 3
**Significance:** 3
**Originality:** 2
**Rating:** 4
**Confidence:** 3

**Summary:**

This paper addresses a key issue in low-rank optimizers for LLM training: the update directions tend to collapse into the same subspace over time, limiting learning. The authors propose a simple method to fix this while keeping the efficiency benefits of low-rank updates.

**Main contributions**

- Propose **SARA**, a sampling method that selects singular vectors based on their importance to keep update subspaces diverse.
- Prove that SARA maintains the convergence rate of standard low-rank methods while preserving memory and compute efficiency.
- Show that SARA can be easily added to existing optimizers like GaLore, Fira, and Adafactor without major code changes.
- Demonstrate improved results on LLaMA models, reducing the gap between low-rank and full-rank Adam on C4 and SlimPajama.
- Provide evidence that SARA improves subspace diversity using overlap and rank-based metrics.

**Questions:**

1. **Why sample by singular values?**
   Why are the sampling probabilities proportional to singular values, for instance, why not proportional to exp(.) of singular values to resemble a softmax function? Have you tried uniform sampling? An ablation would clarify this design choice.

> **I understand that running additional experiments might not be feasible, I'd be happy to increase my score if authors can provide some discussion and maybe run some simple experiments with uniform sampling.**

2. **Aggregation of subspace metrics**
   Section 4.3 shows overlap for only one parameter group. Please aggregate results over all layers and groups (e.g., average or any reasonable summarization metric for overlap per layer and per training step) to give a complete picture.

> **I am willing to increase my score if authors can aggregate the results for subspace metrics.**

3. **How often is SVD computed?**
   How frequently do you compute the SVD and update the sampling probabilities? I could not find details about the hyperparameter $ \tau $ in the paper. I assume it matches the baseline methods, but please clarify and report it in Appendix B. Also, a rough estimate of the added runtime or FLOP overhead would be helpful.

4. **Is sampling stable across runs?**
   Does the added randomness affect training stability or convergence? Showing results across multiple seeds would help. Also, does random sampling affect the optimal value for hyperparameter $\tau$?

5. **Typo in Figure 4?**
   The x-axis in Figure 4 (and similar plots in the appendix) is labeled "update steps," but it appears to show singular value indices. Please clarify or correct the label.

**Ethical Concerns:**

["NO or VERY MINOR ethics concerns only"]

**Final Justification:**

The rebuttal addressed most of my actionable concerns: it added uniform and top-r baselines over three seeds showing consistent PPL gains for importance sampling, clarified SVD refresh ($\tau=200$) with negligible overhead, and reported multi-seed (3 seeds) stability and a $\tau$ sweep (best $\approx 200$). They also averaged subspace overlap across layers in the rebuttal and indicated that figure labeling/organization will be fixed in the revision.

Remaining issues: there is still no comparison to a softmax sampling variant, and the rationale for using probabilities proportional to singular values (rather than other importance-sampling schemes) needs a clearer justification. In addition, the layer-wise aggregation and the new baselines should be included in the main paper with transparent reporting (means, standard deviations or 95% CIs, number of seeds, and training tokens). With these caveats, the rebuttal strengthens the work enough to warrant a borderline accept (4).

**Limitations:**

Add a short paragraph acknowledging technical limits: extra SVD cost, possible instability from random sampling, etc.

**Quality:**

3

**Strengths And Weaknesses:**

## Strengths

- **Clear motivation.** The paper addresses a real issue in low-rank training: the update directions collapse into the same subspace over time, limiting learning. This is a well-motivated and practically important problem.
- **Simple and useful method.** The proposed SARA algorithm is easy to implement, requires no extra memory or compute, and can be applied to many existing optimizers.
- **Theoretical support.** While not central to the paper’s impact, the convergence analysis adds technical credibility.
- **Strong experiments.** The paper includes extensive experiments on LLaMA models, showing consistent improvements and closing much of the gap with full-rank Adam.

## Weaknesses

- **Unorganized experiments.** Section 4.3 visualizes subspace metrics for just one parameter group, with the rest of the figures relegated to the appendix. The paper should aggregate and discuss results across all layers and groups in the main text (for example, plotting the average subspace overlap per layer and the average overlap over training steps for each parameter group) to give a clearer, more comprehensive view of the method’s behavior.
- **No comparison with alternative sampling methods.** The paper does not benchmark SARA against simpler schemes such as uniform, top-k, or random-k sampling, leaving it unclear how much of the gain is due to importance sampling versus just adding noise or diversity.
- **Writing could be improved.** Some parts of the paper are under-explained, especially the sampling procedure. The design choices, such as sampling probabilities being proportional to singular values, are not well motivated or justified.
- **No analysis of randomness.** It is unclear whether the randomness introduced by sampling could hurt performance or stability in some cases. A discussion or ablation would help. For instance, does it affect the distributed training/parallelization?

---

> ### Author Rebuttal · Authors · 2025-07-29
>
> We express our deepest gratitude to the reviewer for the time and effort in reviewing our work. It is very encouraging for us to see your evaluation of our work as 1). having clear motivation, 2). simple and useful, and 3). containing theoretical support and strong experiments. Below, we want to address your concerns about our work.
>
> **Question 1 and Weakness 3**: Why are the sampling probabilities proportional to singular values, for instance, why not proportional to exp(.) of singular values to resemble a softmax function? Have you tried uniform sampling? An ablation would clarify this design choice.
>
>
> **Response to Question 1 and Weakness 3**: The frozen dominant subspace phenomenon was recently formalized in [1, 2], and to the best of our knowledge, our paper is the first to propose a theoretically grounded and empirically validated method to mitigate it. Specifically, while prior methods like GaLore and $Q$-GaLore project gradients onto the dominant subspace (defined by leading singular vectors), we show empirically (e.g., Figure 3, Table 1) that such a strategy can lead to stagnation in subspace evolution during pretraining, ultimately limiting the expressiveness of weight updates.
>
> To tackle this, we introduce an importance sampling mechanism inspired by classical sketching techniques in randomized linear algebra. Given a big data matrix $A \in \mathbb{R}^{n \times d}$, sketching aims to reduce dimensionality by sampling a small matrix $SA$, where $S \in \mathbb{R}^{s \times n}$ and $s \ll n$. A principled way to construct $S$ is by leveraging leverage scores, which quantify each row’s contribution to the row space of $A$. Rather than selecting the top-$k$ leverage rows (which could lead to overfitting to a narrow subspace), sketching theory recommends sampling **proportionally** to leverage scores, which better preserves the full subspace structure [3].
>
> We draw a direct analogy between this and our method: we treat the singular values (quantifying each singular vector’s contribution) of the gradient as analogous to leverage scores, and sample left singular vectors without replacement, **proportional** to their singular values. This avoids always selecting the same top-$k$ directions, like in GaLore and $Q$-GaLore, and instead promotes diversity across training steps. Our importance sampling method is not a heuristic adjustment made purely for empirical performance. Rather, it is a theoretically motivated approach rooted in randomized linear algebra and directly addresses the frozen subspace problem. While LoRA-related work focuses on augmenting low-rank structure, our work focuses on dynamically diversifying it during optimization, which is a crucial distinction.
>
> Moreover, SVD is a well-established and widely used technique in machine learning. Many methods—such as principal component analysis (PCA) [4] and attention approximation techniques [5]—rely on projecting data onto the top-$k$ left singular vectors of a data matrix to reduce dimensionality. Our randomized importance sampling strategy offers a novel perspective: instead of deterministically selecting the top directions, probabilistic selection may enhance subspace diversity and performance. We believe this insight could inspire future work in improving other applications that currently rely on fixed top-$k$ subspace projections and in theoretically and/or empirically studying the similarities and differences between our importance sampling and leverage score sampling.
>
> **Weakness and Question 2**: No comparison with alternative sampling methods. The paper does not benchmark SARA against simpler schemes such as uniform, top-$k$, or random-$k$ sampling, leaving it unclear how much of the gain is due to importance sampling versus just adding noise or diversity.
>
> **Response to Weakness and Question 2**: We run experiments with uniform sampling, our proposed importance sampling, and top-$r$ selection (the subspace selection used in GaLore) using three different seeds on the LLaMA-130M model with a sequence length of 256. The models are trained for 2 billion tokens. The results are provided in the table below. We can see that importance sampling outperforms both uniform sampling and top-$r$ selection.
>
> |   | Seed 1 | Seed 2 | Seed 3 | mean|
> |-----|-----|-----|-----|-----|
> |Top-$r$ | 24.92 | 24.95 | 24.99 | 24.96|
> |Uniform Sampling | 24.69 | 24.68 | 24.69 | 24.68|
> |Importance Sampling | 24.46 | 24.47 | 24.54 | 24.49|
>
>
>
> We average the subspace metrics across all layers, and the resulting averages are presented in the table below. We observe that the overlap between the anchor subspace and subsequent subspaces is lower for GaLore-SARA-Adam compared to GaLore-Adam. This indicates that SARA encourages the optimization trajectory to explore a more diverse set of subspaces relative to using the dominant subspace. The averaged anchor subspace overlap metrics for GaLore-Adam and GaLore-SARA-Adam across all layers are summarized as follows:
>
> |Training iterations | 2200 | 2400 | 2600 | 2800 | 3000 | 3200 | 3400 | 3600 | 3800 | 4000|
> |-----|-----|-----|-----|-----|-----|-----|-----|-----|-----|-----|
> |GaLore-Adam | 0.6328 | 0.6328| 0.6328| 0.6289| 0.6289| 0.6211| 0.6211| 0.6172| 0.6133 | 0.6094|
> |GaLore-SARA-Adam | 0.3516| 0.3535| 0.3496| 0.3496| 0.3477| 0.3477| 0.3477| 0.3438| 0.3418| 0.3398|
>
>
> **Question 3**: How frequently do you compute the SVD and update the sampling probabilities? I could not find details about the hyperparameter $\tau$ in the paper. I assume it matches the baseline methods, but please clarify and report it in Appendix B. Also, a rough estimate of the added runtime or FLOP overhead would be helpful.
>
> **Response to Question 3**: SVD is computed every $\tau$ iterations, where $\tau$ is the subspace update period of our algorithm. In our experiments, we set $\tau$ to the same value as in GaLore’s original paper, which is 200. We have included it in the updated version of our paper.
>
> The additional overhead introduced by sampling is minimal. To address your concern, we conducted an additional experiment. In the experiment, we measure the wall-clock time of SVD computation and sampling. We calculate the average wall-clock time of ten runs, the result for SVD is 0.34 seconds for a 2048 by 2048 matrix, and the result for sampling is 0.0005 seconds. The empirical result shows that the addition computational cost brought by sampling is negligible.
>
>
> **Weakness 4 and Question 4**: Does the added randomness affect training stability or convergence? Showing results across multiple seeds would help. Also, does random sampling affect the optimal value for the hyperparameter $\tau$?
>
> **Response to Weakness 4 and Question 4**: Sampling is stable across runs. In our **Response to Weakness and Question 2**, we show the results of GaLore-SARA-Adam across three seeds, demonstrating its stability. We also test whether random sampling affects the optimal subspace update frequency, i.e., $\tau$. Specifically, we train the LLaMA-130M model with different $\tau$ values on the C4 dataset with a sequence length of 256. The results are shown in the table below. We find that the optimal $\tau$ remains 200. This indicates that random sampling does not affect the optimal value of $\tau$, which is an important hyperparameter.
>
> | $\tau$ | 100 | 200 | 300 | 400 | 500 |
> |-----|-----|-----|-----|-----|-----|
> |Evaluation PPL | 24.78 | 24.46 | 24.51 | 24.78 | 24.82|
>
> Also, in our experiment, we did not find that sampling has any negative effect on distributed training.
>
> **Weakness 1 and Question 5**: Organization and typos. Section 4.3 visualizes subspace metrics for just one parameter group, with the rest of the figures relegated to the appendix. The paper should aggregate and discuss results across all layers and groups in the main text (for example, plotting the average subspace overlap per layer and the average overlap over training steps for each parameter group) to give a clearer, more comprehensive view of the method’s behavior.
> The $x$-axis in Figure 4 (and similar plots in the appendix) is labeled "update steps," but it appears to show singular value indices. Please clarify or correct the label.
>
>
> **Response to Weakness 1 and Question 5**: We sincerely appreciate your valuable comments on the organization and typos in our work. We completely agree with your valuable comments, and we have fixed these issues in the revised version of our paper.
>
>
> Once again, we are deeply grateful for your insightful comments. We have addressed all identified weaknesses and questions, and we respectfully and genuinely hope you will consider raising your score.
>
>
> [1] Zhenyu Zhang, Ajay Jaiswal, Lu Yin, Shiwei Liu, Jiawei Zhao, Yuandong Tian, and Zhangyang Wang. "Q-galore: Quantized galore with int4 projection and layer-adaptive low-rank gradients." Preprint'24.
>
> [2] Guy Gur-Ari, Daniel A. Roberts, and Ethan Dyer. "Gradient descent happens in a tiny subspace." Preprint'18.
>
> [3] Michael W. Mahoney "Randomized algorithms for matrices and data." Foundations and Trends in Machine Learning'11.
>
> [4] Jianqing Fan, Qiang Sun, Wen-Xin Zhou, and Ziwei Zhu. "Principal component analysis for big data." Preprint'18.
>
> [5] Sinong Wang, Belinda Z. Li, Madian Khabsa, Han Fang, and Hao Ma. "Linformer: Self-attention with linear complexity." Preprint'20.

---

> > ### Comment · Reviewer_yFMY · 2025-08-02
> >
> > Thank you for the additional results and clarifications. I am raising my score to 4 (borderline accept).
> >
> > That said, the paper could be significantly improved by including more sampling baselines and ablations in the main text, as well as providing a clearer sensitivity analysis of the $\tau$ hyperparameter to better support its claims and comparisons.

---

> > > ### Author Response · Authors · 2025-08-02
> > > **Thank you very much!**
> > >
> > > We sincerely thank the reviewer for taking the time to read our additional results and clarification. We are committed to incorporating your valuable suggestions into the final version of the paper.

---

### Official Review · Reviewer_CjAj · 2025-07-03

**Clarity:** 4
**Significance:** 3
**Originality:** 3
**Rating:** 5
**Confidence:** 3

**Summary:**

The paper tackles a limitation of recent low-rank optimizers such as GaLore and Fira: after a few thousand iterations the dominant gradient subspace stabilises, so subsequent updates accumulate in almost the same low-rank slice and stall expressiveness.

To break this "frozen subspace", the authors introduce SARA—Importance SAmpling for Low-RAnk optimisation. Every $\tau$ steps they compute the SVD of the minibatch gradient and sample r left singular vectors without replacement in proportion to their singular values (Alg. 2). The resulting projector replaces the usual top-r basis and can be plugged into any gradient-projection optimiser.

Key claims are:

* Theory – SARA's stochastic projector yields an $O(L\Delta/(\delta^{2.5}T)+\sqrt{L\Delta}\sigma/(\delta^{3.5}T))$ convergence rate (Theorem 3.4), matching the expected $O(1/\sqrt{T})$ for SGD momentum in non-convex smooth objective. In comparison, there is no convergence guarantees for the vanilla GaLore method.

* Practice – On LLaMA-60 M/130 M/350 M/1.1 B pre-training over C4 and SlimPajama, SARA paired with GaLore or Fira achieves slightly better or on-par performance when compared with Galore in terms of perplexity.

* Dynamics – SARA lowers adjacent-subspace overlap (Figure 1, 3) and produces higher-rank weight updates (Figure 4), supporting the motivation.

**Questions:**

* What is the performance of SARA in larger-sized models, such as Llama-3.1-8B?

* Do the improvements in subspace diversity lead to any performance gains for downstream tasks, such as math, commonsense reasoning, or instruction-following?

* I am wondering why the similarity between adjacent projection matrices is large, given the diversity in sampled batches, and the noise is expected to be a dominant term in the gradient in the small batch training setting?

**Ethical Concerns:**

["NO or VERY MINOR ethics concerns only"]

**Final Justification:**

The rebuttal addresses most of my concerns, and I would like to maintain my current rating.

**Limitations:**

Although the theoretical analysis is conducted under SGD with momentum, the experimental results are based on the Adam optimizer. Given the challenges in establishing convergence guarantees for Adam, it would be helpful to explicitly acknowledge this discrepancy in the limitations section (e.g., Appendix C).

**Paper Formatting Concerns:**

No major formatting issues in this paper are observed.

**Quality:**

3

**Strengths And Weaknesses:**

### Strength

1. Applying sampling to subspace selection in low-rank projection methods is novel, and compelling to me.

2. The theoretical analysis attains the convergence rate expected for momentum-based SGD on smooth objectives, and addresses the non-convergence issue of Galore.

3. The paper is clearly written and straightforward to follow.

4. The experimental results are comprehensive and convincingly support the authors' claims.

### Weakness

1. **Scalability**: Adding experiments on larger language models—for instance, Llama-3.1-8B—would strengthen the empirical evidence for the method’s scalability.

2. **Theory-practice Gap**: While the theorem is proven for SGD momentum, the experiments use Adam for optimization. I understand that it is difficult to prove convergence guarantees for Adam given its non-projection-free property, but it would be beneficial to mention this in the limitations (Appendix C).

3. **Minor presentation issues**:

* Some missing citations [1–4].

* Algorithm 1, line 7: The formatting could be refined for readability.

* Tables 1–4: Sorting the results in descending order would make the comparisons clearer.

* Line 469: The statement should involve inequalities rather than equalities.

### References

[1]: Liu, Xinxin, et al. "Refining Salience-Aware Sparse Fine-Tuning Strategies for Language Models." arXiv preprint arXiv:2412.13488 (2024).

[2]: Pan, Rui, et al. "LISA: layerwise importance sampling for memory-efficient large language model fine-tuning." NeurIPS 2024.

[3]: Hui, Tingfeng, et al. "Hft: Half fine-tuning for large language models." arXiv preprint arXiv:2404.18466 (2024).

[4]: Li, Pengxiang, et al. "Owlore: Outlier-weighed layerwise sampled low-rank projection for memory-efficient llm fine-tuning." arXiv preprint arXiv:2405.18380 (2024).

---

> ### Author Rebuttal · Authors · 2025-07-29
>
> We express our deepest gratitude to the reviewer for the time and effort in reviewing our work. It is very encouraging for us to see your evaluation of our work as 1). novel and compelling, 2). clearly written, and 3). containing sufficient theoretical and experimental results. Below, we want to address your concerns about our paper.
>
> **Weakness and Question 1**: Adding experiments on larger language models—for instance, Llama-3.1-8B—would strengthen the empirical evidence for the method’s scalability.
>
> **Response to Weakness and Question 1**: Thank you very much for your insightful comments. We currently report experimental results on models with 60M, 130M, 350M, and 1.1B parameters (Tables 1 and 2), demonstrating that SARA scales effectively across a range of model sizes. Due to limitations in time and computational resources, it is very challenging to present results on larger models at this stage. Even conducting experiments on the 1.1B model takes more than 10 days by using all of our GPUs. Nevertheless, we would like to emphasize that our theoretical convergence guarantee (Theorem 3.4) is independent of model size and thus supports general scalability.
>
> Although we lack the time and resources to run very large-scale experiments, we have fully utilized the rebuttal period to present the following experimental results that are computationally efficient and feasible within this limited timeframe.
>
> $\bullet$ **Hyperparameter sensitivity**: Demonstrated stable performance across critical hyperparameters (see our **Response to Weakness and Question 2** to **Reviewer ZP7Q**).
>
> $\bullet$ **Additional metrics beyond validation perplexity**: Confirmed enhanced downstream task performance (see our **Response to Weakness and Question 1** to **Reviewer ZP7Q**).
>
> $\bullet$ **Comparison with uniform sampling**: Highlighted advantages of our sampling strategy (see our **Response to Weakness and Question 2** to **Reviewer yFMY**).
>
> $\bullet$ **Multiple random seeds**: Verified the robustness and reliability of our method across different runs (see our **Response to Weakness and Question 2** to **Reviewer yFMY**).
>
> **Weakness 2**: While the theorem is proven for SGD momentum, the experiments use Adam for optimization. I understand that it is difficult to prove convergence guarantees for Adam given its non-projection-free property, but it would be beneficial to mention this in the limitations (Appendix C).
>
> **Response to Weakness 2**: We thank you for your insightful comments. We use Adam since it is a widespread adoption in large-scale LLM pretraining for its empirical performance. We completely agree with your insightful comments that proving convergence guarantees for Adam, especially under low-rank projection with importance sampling, remains challenging due to its complex dynamics and lack of a projection-free formulation. We have included this discussion in the limitations part of our paper. We thank you again for pointing this out!
>
> **Weakness 3**: Some missing citations [1–4]. Algorithm 1, line 7: The formatting could be refined for readability. Tables 1–4: Sorting the results in descending order would make the comparisons clearer. Line 469: The statement should involve inequalities rather than equalities.
>
>
> **Response to Weakness 3**: Thank you very much for pointing out these wonderful papers [1–4]. We have carefully read these recent works, and we believe they are highly related to our work. [1] analyzes the sparsity-based parameter-efficient fine-tuning (SPEFT) for LLMs, which is an alternative method of LoRA. [2] designs a novel method for memory-efficient fine-tuning for LLMs. It has been shown that it can outperform LoRA and full-parameter training in many cases. Similarly, [3] proposes another novel fine-tuning method called Half Fine-Tuning (HFT), which can mitigate "catastrophic forgetting" in LLMs during sequential training and instruction tuning. Finally, [4] also proposes a novel memory-efficient fine-tuning method by strategically selecting layers to update based on outlier statistics. Our paper also considers the memory and convergence behavior, but the difference is that we focus on the LLM pretraining instead of fine-tuning. We have included the above discussion in the revised version of our paper.
>
> Regarding Algorithm 1 line 7, we have updated it as the following two lines:
> \begin{align*}
>     \mathcal{S} &\gets \\{V_l^{(t - 1)}, M_l^{(t - 1)}, x_l^{(t)}, P_l^{(t)}, G_l^{(t)}, \beta_1, \beta_2, \xi, \eta, \alpha\\}. ~~~~~~~~~\text{Input parameters}\\\\
>     x_l^{(t)} &\gets \text{GaLore-Adam}(\mathcal{S}) \text{ or } \text{Fira-Adam}(\mathcal{S}).
> \end{align*}
> Thank you for your insightful comments.
>
> Also, we have fixed our Tables 1–4 to make "the results in descending order" to make them clearer.
>
> Finally, regarding Line 469, we first want to sincerely thank you for carefully checking the details of our work, even though this is in the appendix. In Line 469, we have:
> \begin{align*}
> \mathbb{E} \left[ \left\\| \tilde{M}_l^{(0)} - \nabla_l f(x^{(0)}) \right\\|_F^2 \right]
> &= \mathbb{E} \left[ \left\\| \beta_1 P_l^{(0)} (P_l^{(0)})^\top \left( G_l^{(0)} - \nabla_l f(x^{(0)}) \right) \right\\|_F^2 \right] \\\\
> &\quad + \mathbb{E} \left[ \left\\| \left( \beta_1 P_l^{(0)} (P_l^{(0)})^\top - I \right) \nabla_l f(x^{(0)}) \right\\|_F^2 \right].
> \end{align*}
> This is because in general $\mathbb{E}[\\|A + B\\|_F^2] = \mathbb{E}[\\|A\\|_F^2] + \mathbb{E}[\\|B\\|_F^2] + 2\mathbb{E}[\langle A, B \rangle]$, but here, we have $\mathbb{E}[\langle A, B \rangle] = 0$, where $A = \beta_1 P_l^{(0)} (P_l^{(0)})^\top \left( G_l^{(0)} - \nabla_l f(x^{(0)}) \right)$ and $B = \left( \beta_1 P_l^{(0)} (P_l^{(0)})^\top - I \right) \nabla_l f(x^{(0)})$. The reason is that $A$ has ``zero-mean'' since we have $G_l^{(0)}$ is unbiased, as our assumption.
>
> **Question 2**: Do the improvements in subspace diversity lead to any performance gains for downstream tasks, such as math, commonsense reasoning, or instruction-following?
>
> **Response to Question 2**: Thank you for your insightful comments. We have conducted additional experiments using multiple downstream tasks from the GLUE benchmark to further evaluate the effectiveness of our method. The table below compares fine-tuning performance on GLUE tasks using pre-trained RoBERTa-Base with different optimizers and LoRA configurations. Results for LoRA and GaLore-Adam are from Zhao et al. [7]. GaLore-SARA-Adam uses identical hyperparameters to GaLore-Adam for the fairness of comparison.
>
> | | CoLA | STS-B | MRPC | RTE | SST2 | MNLI | QNLI | QQP | Average |
> |-----|-----|-----|-----|-----|-----|-----|-----|-----|-----|
> |LoRA (rank 4) | 61.38 | 90.57 | 91.07 | 78.70 | 92.89 | 86.82 | 92.18 | 91.29 | 85.61 |
> |GaLore-Adam (rank 4) | 60.35 | 90.73 | 92.25 | 79.42 | 94.04 | 87.00 | 92.24 | 91.06 | 85.89 |
> |GaLore-SARA-Adam ( rank 4) | 61.34 | 90.81 | 91.17 | 80.50 | 94.15 | 87.48 | 92.73 | 91.27 | 86.18|
> |LoRA (rank 8) | 62.83 | 90.80| 91.90 | 79.06 | 93.46 | 86.94 | 92.25 | 91.22 | 85.93|
> |GaLore-Adam (rank 8) | 60.06 | 90.82 | 92.01 | 79.78 | 94.38 | 87.17 | 92.20 | 91.11 | 85.94|
> |GaLore-SARA-Adam (rank 8) | 61.65 | 90.72 | 92.90 | 78.70 | 94.15 | 87.52 | 92.91 | 91.51 | 86.25|
>
> **Question 3**: I am wondering why the similarity between adjacent projection matrices is large, given the diversity in sampled batches, and the noise is expected to be a dominant term in the gradient in the small batch training setting?
>
> **Response to Question 3**: The randomness in mini-batch gradients can indeed introduce some variation in the dominant subspaces across steps. However, this typically does not have a significant impact on the similarity between adjacent projection matrices, as demonstrated empirically in [5]. In the problem setting of [6], the subspace spanned by the top-$k$ singular vectors tends to stabilize early in training and remains nearly fixed, even though individual singular vectors may vary. The gradient dynamically adapts to reside within this top subspace, resulting in a consistent projection structure across steps [6].
>
> We sincerely thank you for raising this important point. We have incorporated this clarification into the final version of our paper, as we believe it strengthens the motivation behind our work.
>
> Once again, we are deeply grateful for your insightful comments. We have addressed all identified weaknesses and questions, and we respectfully and genuinely hope you will consider raising your score.
>
>
> [1] Xinxin Liu, Aaron Thomas, Cheng Zhang, Jianyi Cheng, Yiren Zhao, and Xitong Gao. "Refining Salience-Aware Sparse Fine-Tuning Strategies for Language Models." ACL'25.
>
> [2] Rui Pan, Xiang Liu, Shizhe Diao, Renjie Pi, Jipeng Zhang, Chi Han, and Tong Zhang. "Lisa: Layerwise importance sampling for memory-efficient large language model fine-tuning." NeurIPS'24.
>
> [3] Tingfeng Hui, Zhenyu Zhang, Shuohuan Wang, Weiran Xu, Yu Sun, and Hua Wu. "Hft: Half fine-tuning for large language models." Preprint'24.
>
> [4] Pengxiang Li, Lu Yin, Xiaowei Gao, and Shiwei Liu. "Owlore: Outlier-weighed layerwise sampled low-rank projection for memory-efficient llm fine-tuning." Preprint'24.
>
> [5] Zhenyu Zhang, Ajay Jaiswal, Lu Yin, Shiwei Liu, Jiawei Zhao, Yuandong Tian, and Zhangyang Wang. "Q-galore: Quantized galore with int4 projection and layer-adaptive low-rank gradients." Preprint'24.
>
> [6] Guy Gur-Ari, Daniel A. Roberts, and Ethan Dyer. "Gradient descent happens in a tiny subspace." Preprint'18.
>
> [7] Jiawei Zhao, Zhenyu Zhang, Beidi Chen, Zhangyang Wang, Anima Anandkumar, and Yuandong Tian. "Galore: Memory-efficient llm training by gradient low-rank projection." ICML'24.

---

### Official Review · Reviewer_ZP7Q · 2025-07-03

**Clarity:** 2
**Significance:** 3
**Originality:** 3
**Rating:** 3
**Confidence:** 2

**Summary:**

This paper investigates the issue of "frozen" dominant subspaces in low-rank optimization during LLM pretraining, demonstrating that conventional dominant subspace selection leads to highly overlapping and restrictive update directions. To address this, the authors introduce SARA, i.e., importance sampling for low rank optimization, which randomly samples singular vectors weighted by their importance to construct more diverse update subspaces. Theoretical convergence guarantees are presented and SARA is shown empirically to outperform prior low-rank optimization approaches across several LLM sizes and optimizers, demonstrating reduced perplexity gaps versus full-rank baselines.

**Questions:**

Please see the weaknesses. In general:

1.Beside validation perplexity, more metrics are needed to show the model performance.

2.More analysis of hyperparameter sensitivity is needed.

3.It's better to scale up the model size to prove the generalization ability of the method.

**Ethical Concerns:**

["NO or VERY MINOR ethics concerns only"]

**Limitations:**

Yes.

**Paper Formatting Concerns:**

No.

**Quality:**

3

**Strengths And Weaknesses:**

Strengths

1.The paper provides a clear and well-motivated analysis of the limitations of dominant subspace selection in existing low-rank optimization methods, supported by empirical observations (see Figure 2) that show increasing and later-stable overlap of dominant subspaces during pretraining.

2.SARA is a simple yet principled modification that introduces importance sampling into the subspace selection process, which reduces overlap between adjacent subspaces (evident in Figure 3) and encourages more effective exploration without adding computational overhead.

3.The methodology is well-integrated with existing low-rank optimizers (e.g., GaLore and Fira), and the plug-and-play nature is likely to make adoption easy within the community.

Weaknesses

1.The evaluation focuses mainly on perplexity during pretraining. While this is the standard criterion for language modeling, it is not evident how SARA’s purported improvements translate into downstream performance (such as finetuning, alignment, or other application-level metrics). The practical significance of reduced PPL gap, especially at scale, is not elaborated on beyond pretraining.

2.While the method is said to be plug-and-play and robust, there is little discussion or systematic ablation regarding the impact of SARA’s hyperparameters (e.g., the subspace refresh frequency τ, the sampled rank r, the total number of candidate singular vectors m) on convergence and final perplexity. This leaves uncertainty on how sensitive SARA is to these settings and may make reproducibility and deployment less straightforward.

3.While the experiments include the model ranging from 60M to 1.1B, there is still a gap between this and the current mainstream LLM size, such as 7B or 32B.  Considering the problem of training cost, if the loss/perplexity changing chart and table under less training tokens can be provided, the paper can also be made more solid.

---

> ### Author Rebuttal · Authors · 2025-07-29
>
> We express our deepest gratitude to the reviewer for the time and effort in reviewing our work. It is very encouraging for us to see your evaluation of our work as 1). clear and well-motivated, 2). simple and has principled modification, and 3). well-integrated with existing low-rank optimizers. Below, we want to address your concerns about our work.
>
> **Weakness and Question 1**: Besides validation perplexity, more metrics are needed to show the model's performance.
>
> **Response to Weakness and Question 1**: Thank you for your insightful comments. We have conducted additional experiments using multiple downstream tasks from the GLUE benchmark to further evaluate the effectiveness of our method. The table below compares fine-tuning performance on GLUE tasks using pre-trained RoBERTa-Base with different optimizers and LoRA configurations. Results for LoRA and GaLore-Adam are from Zhao et al. [1]. GaLore-SARA-Adam uses identical hyperparameters to GaLore-Adam for the fairness of comparison.
>
> | | CoLA | STS-B | MRPC | RTE | SST2 | MNLI | QNLI | QQP | Average |
> |-----|-----|-----|-----|-----|-----|-----|-----|-----|-----|
> |LoRA (rank 4) | 61.38 | 90.57 | 91.07 | 78.70 | 92.89 | 86.82 | 92.18 | 91.29 | 85.61 |
> |GaLore-Adam (rank 4) | 60.35 | 90.73 | 92.25 | 79.42 | 94.04 | 87.00 | 92.24 | 91.06 | 85.89 |
> |GaLore-SARA-Adam ( rank 4) | 61.34 | 90.81 | 91.17 | 80.50 | 94.15 | 87.48 | 92.73 | 91.27 | 86.18|
> |LoRA (rank 8) | 62.83 | 90.80| 91.90 | 79.06 | 93.46 | 86.94 | 92.25 | 91.22 | 85.93|
> |GaLore-Adam (rank 8) | 60.06 | 90.82 | 92.01 | 79.78 | 94.38 | 87.17 | 92.20 | 91.11 | 85.94|
> |GaLore-SARA-Adam (rank 8) | 61.65 | 90.72 | 92.90 | 78.70 | 94.15 | 87.52 | 92.91 | 91.51 | 86.25|
>
>
>
> **Weakness and Question 2**: More analysis of hyperparameter sensitivity is needed.
>
> **Response to Weakness and Question 2**: We thank you for pointing this out. We have performed additional sensitivity analysis on the subspace refresh frequency $\tau$ and the sampled rank $r$ by training a LLaMA-130M model on the C4 dataset with sequence length 256. The sensitivity of performance to subspace refresh frequency $\tau$ is presented in the table below, with an optimal setting around $\tau=200$. Additionally, when $\tau$ varies from 100 to 500, our algorithm's performance remains stable:
>
>
> | $\tau$ | 100 | 200 | 300 | 400 | 500 |
> |-----|-----|-----|-----|-----|-----|
> |Evaluation PPL | 24.78 | 24.46 | 24.51 | 24.78 | 24.82|
>
> The impact of the sampled rank $r$ on perplexity is demonstrated in the table below, confirming our intuition that a higher rank consistently yields lower evaluation perplexity:
>
> | rank $r$ | 100 | 200 | 300 | 400 |
> |-----|-----|-----|-----|-----|
> |Evaluation PPL | 27.70 | 25.23 | 24.04 | 23.29 |
>
>
> **Weakness and Question 3**: It’s better to scale up the model size to prove the generalization ability of the method.
>
> **Response to Weakness and Question 3**: Thank you very much for your insightful comments. We currently report experimental results on models with 60M, 130M, 350M, and 1.1B parameters (Tables 1 and 2), demonstrating that SARA scales effectively across a range of model sizes. Due to limitations in time and computational resources, it is very challenging to present results on larger models at this stage. Even conducting experiments on the 1.1B model takes more than 10 days by using all of our GPUs. Nevertheless, we would like to emphasize that our theoretical convergence guarantee (Theorem 3.4) is independent of model size and thus supports general scalability.
>
> Although we lack the time and resources to run very large-scale experiments, we have fully utilized the rebuttal period to present the following experimental results that are computationally efficient and feasible within this limited timeframe.
>
> $\bullet$ **Hyperparameter sensitivity**: Demonstrated stable performance across critical hyperparameters (see our **Response to Weakness and Question 2** to **Reviewer ZP7Q**).
>
> $\bullet$ **Additional metrics beyond validation perplexity**: Confirmed enhanced downstream task performance (see our **Response to Weakness and Question 1** to **Reviewer ZP7Q**).
>
> $\bullet$ **Comparison with uniform sampling**: Highlighted advantages of our sampling strategy (see our **Response to Weakness and Question 2** to **Reviewer yFMY**).
>
> $\bullet$ **Multiple random seeds**: Verified the robustness and reliability of our method across different runs (see our **Response to Weakness and Question 2** to **Reviewer yFMY**).
>
> Once again, we are deeply grateful for your insightful comments. We have addressed all identified weaknesses and questions, and we respectfully and genuinely hope you will consider raising your score.
>
> [1] Jiawei Zhao, Zhenyu Zhang, Beidi Chen, Zhangyang Wang, Anima Anandkumar, and Yuandong Tian. "Galore: Memory-efficient llm training by gradient low-rank projection." ICML'24.

---

> > ### Author Response · Authors · 2025-08-06
> > **A Kind Reminder: Discussion Period Will End Soon**
> >
> > Again, we would like to express our deepest gratitude for the time and effort that you have dedicated to our manuscript, as well as for your insightful comments. We kindly remind you that the discussion period will end soon (Aug 8, 11.59pm AoE). We have carefully considered all feedback from you and other reviewers and believe that we have responded to each comment. We look forward to receiving further valuable follow-up comments from you before the discussion period ends. Should there be any further concerns, we remain eager to engage in additional discussion with you.
> >
> > Below, for your reference, we summarize the main strengths and weaknesses of our work as assessed by you and other reviewers, along with our responses to these weaknesses.
> >
> > ## Strengths
> >
> > Reviewers highlighted several strengths of our paper:
> >
> > $\bullet$ Our paper has a clear motivation and problem identification, namely the “frozen dominant subspace” problem in low-rank optimization for LLM pretraining (Reviewers ZP7Q, yFMY, and TBg6).
> >
> > $\bullet$ Our proposed method SARA is straightforward and easy to implement (Reviewers ZP7Q, CjAj, yFMY, and TBg6).
> >
> > $\bullet$ We provide formal theoretical support (Reviewers CjAj, yFMY, and TBg6) as well as comprehensive experimental results (Reviewers ZP7Q, CjAj, and yFMY).
> >
> > $\bullet$ Finally, our paper is clearly written and easy to understand (Reviewers ZP7Q, CjAj, and TBg6).
> >
> >
> > ## Weaknesses and Rebuttals
> >
> > However, the reviewers also pointed out some weaknesses, and we have responded to all of them. These weaknesses and rebuttals can be summarized as follows:
> >
> > $\bullet$ Reviewers ZP7Q, TBg6, and CjAj raised questions about the performance of SARA on criteria and downstream tasks beyond perplexity. In our rebuttal, we conducted additional experiments using multiple downstream tasks from the GLUE benchmark to further evaluate the effectiveness of our method.
> >
> > $\bullet$ Reviewers ZP7Q, TBg6, and yFMY expressed concerns regarding the impact of SARA’s hyperparameters, the effect of randomness on training stability, and comparisons with alternative sampling methods. In response, we performed additional experiments demonstrating that our algorithm is stable across different choices of the hyperparameters $\tau$ and $r$. We also compared our proposed importance sampling with uniform sampling and top-$r$ selection across three different random seeds.
> >
> > $\bullet$ Reviewers yFMY and TBg6 inquired why we sample probabilities proportional to singular values and what contributions our work offers beyond analyzing low-rank subspaces. To address these, we emphasize that, to the best of our knowledge, our work is the first to analyze—both theoretically and empirically—the newly identified challenge of the frozen dominant subspace phenomenon within this paradigm. Our technique of sampling probabilities proportional to singular values is inspired by proportional sampling in leverage score methods from sketching theory, and we are the first to extend this idea to sampling singular values in SVD.
> >
> > $\bullet$ Beyond these points, we have also addressed concerns about the practical significance of the improvements, empirically validated that SARA “does not bring extra overhead,” discussed the frequency of computing SVD, clarified theoretical details (e.g., Line 469), and more. The only concern we cannot immediately address during this rebuttal and discussion period is the generalizability to larger models: due to computational resource limitations, even running experiments on the 1.1B model requires more than 10 days using all our GPUs.
> >
> >
> > ## Conclusion
> >
> > We are confident that we have addressed all major concerns from you and other reviewers and eagerly anticipate your valuable responses. We remain fully committed to revising our manuscript further to ensure its suitability for NeurIPS.

---

> > > ### Comment · Area_Chair_4HdA · 2025-08-08
> > >
> > > Dear Reviewer,
> > >
> > > Could you please reply to the author's response and potentially update your rating? Thank you.
> > >
> > > Best,
> > > AC

---

### Comment · Area_Chair_4HdA · 2025-08-02
**Kindly Engage with Author Responses**

Dear Reviewers,

The authors have submitted their responses to your reviews. At your earliest convenience, please take a moment to engage with their replies. Your continued discussion and clarifications will be invaluable in ensuring a fair and constructive review process for all parties.

Thank you again for your thoughtful contributions and dedication!

Warm regards,

Your AC

---

### Decision · Program_Chairs · 2025-09-17

**Decision:**

Accept (poster)

**Comment:**

The paper gets a borderline rating. For the reviewers who gave negative rating, one didn't participate in the discussion actively. After taking a look at the authors' response, I think most of the reviewers' concern have been addressed reasonably. The AC agrees with Reviewer TBg6 that more experiments on larger models would make the paper stronger, but the AC also thinks that the paper already reaches good quality for publication. The AC read through the manuscript, all reviews, the discussion, and the rebuttal. The authors are highly encouraged to improve the paper quality according to the reviewers' feedback in the camera-ready version. The AC decided to accept this submission.